# Switchable tribology of ferroelectrics

Seongwoo Cho [1,2] ✉, Iaroslav Gaponenko [2,3], Kumara Cordero-Edwards[2], Jordi Barceló-Mercader[4], Irene Arias [4,5], Daeho Kim[1], Céline Lichtensteiger [2], Jiwon Yeom[1], Loïc Musy [2], Hyunji Kim[1], Seung Min Han[1], Gustau Catalan [6,7], Patrycja Paruch[2] ✉ & Seungbum Hong [1,8] ✉

Switchable tribological properties of ferroelectrics offer an alternative route to visualize and control ferroelectric domains. Here, we observe the switchable friction and wear behavior of ferroelectrics using a nanoscale scanning probe–down domains have lower friction coefficients and show slower wear rates than up domains and can be used as smart masks. This asymmetry is enabled by flexoelectrically coupled polarization in the up and down domains under a sufficiently high contact force. Moreover, we determine that this polarization-sensitive tribological asymmetry is widely applicable across various ferroelectrics with different chemical compositions and crystalline symmetry. Finally, using this switchable tribology and multi-pass patterning with a domain-based dynamic smart mask, we demonstrate three-dimensional nanostructuring exploiting the asymmetric wear rates of up and down domains, which can, furthermore, be scaled up to technologically relevant (mm–cm) size. These findings demonstrate that ferroelectrics are electrically tunable tribological materials at the nanoscale for versatile applications.

Ferroelectrics, in which switchable electric polarization is coupled with mechanical deformation, have been extensively studied in view of their technological applications as sensors, actuators, energy harvesters, and memory devices[1–4]. Fundamentally, these materials exhibit electromechanically coupled properties, including piezoelectricity[1], flexoelectricity[5], and electrostriction[6], whose interplay can open critical opportunities in the field of condensed matter physics and functional materials engineering, but is only now beginning to be understood and controlled[7–11]. In addition, oppositely polarized ferroelectric surfaces present different mechanical responses under inhomogeneous deformation as a consequence of the interaction between flexoelectricity and piezoelectricity[12], which could be harnessed for mechanical reading of ferroelectric polarization[13]. As we show in this article, increasing this flexoelectric contribution under highly inhomogeneous stress also has emergent consequences for

coupled tribological properties–in particular, friction and wear behavior–which in turn can be exploited for direct visualization of ferroelectric domains and fine physical lithography without the need for masks or chemical reagents.

Precise control of ferroelectric surface morphology as well as domain structures is essential for numerous applications such as data storage[3,4] and electro-optic devices[14]. During the nanostructuring of materials, including ferroelectrics, patterns with desired size, shape, and periodicity are transferred to the target substrate, generally via an intermediate bridging process using masking, resist, imprint or local thermochemical interactions[15–18]. In contrast, intrinsic properties of the substrate, such as ferroelectric polarization, are rarely employed as a marker for patterning, although we note the demonstration of selective deposition of functionalized nanoparticles[19] and chemical reaction rate difference[20] depending on the surface chemistry of ferroelectric

[1]Department of Materials Science and Engineering, Korea Advanced Institute of Science and Technology (KAIST), Daejeon 34141, Republic of Korea. [2]Department of Quantum Matter Physics, University of Geneva, 1211 Geneva, Switzerland. [3]G.W. Woodruff School of Mechanical Engineering, Georgia Institute of Technology, Atlanta, Georgia 30332, United States of America. [4]LaCàN - Mathematical and Computational Modeling, Polytechnic University of Catalonia, Barcelona 08034, Spain. [5]International Centre for Numerical Methods in Engineering (CIMNE), Barcelona 08034, Spain. [6]Catalan Institute of Nanoscience and Nanotechnology (ICN2), Campus Autonomous University of Barcelona, Bellaterra 08193, Spain. [7]Catalan Institution for Research and Advanced Studies (ICREA), Barcelona 08010, Catalonia. [8]KAIST Institute for NanoCentury (KINC), Korea Advanced Institute of Science and Technology (KAIST), Daejeon 34141, Republic of Korea. ✉e-mail: seongwoo.cho@unige.ch; patrycja.paruch@unige.ch; seungbum@kaist.ac.kr

domains. Nevertheless, there are no reports in which ferroelectric polarization is used as a smart mask for nanoscale patterning based on differential and locally switchable mechanical wear rates, as opposed to chemical etch rates.

In this study, we establish that such switchable, polarization-dependent mechanical lithography is, in fact, possible. We demonstrate that the local friction and wear behavior of ferroelectrics is asymmetric and the existence of the asymmetry is independent of surface chemistry under large strain gradient, and that this inherent tribological asymmetry enables facile and reversible control of friction and wear properties, which can be exploited for nano-lithographic patterning by simply rubbing the surface of a voltage-written ferroelectric. To prove our idea, we start from uniaxial ferroelectric LiNbO$_3$ single crystals as a simple and accessible model system, already widely used in electro-optic applications[21]. We discover flexoelectrically coupled polarization-dependent asymmetric friction and surface wear in these materials by applying a sufficiently high mechanical force using a diamond atomic force microscopy (AFM) probe at the nanoscale or silica nanoparticles at the bulk scale. Furthermore, we confirm that this asymmetry does not originate from either electrostatic effects or inhomogeneous defect distribution but is linked to the competing vs. synergistic interplay of flexoelectric and ferroelectric polarization in oppositely oriented domains—the strain gradients of a downward-pressing tip result in a flexoelectric component which will either oppose or add to the underlying ferroelectric polarization. Switching the ferroelectric domains by local electric field application should thus allow simultaneous and reversible control of the tribological responses (friction and wear) of the material, which in turn can be used to dynamically manipulate surface morphology nanostructures. We extend our findings to LiNbO$_3$ and PbTiO$_3$ thin films and Pb(Zn$_{1/3}$Nb$_{2/3}$)O$_3$–PbTiO$_3$ (PZN-PT) single crystals to establish the wider applicability of the observed tribological asymmetry, allowing for more precise polarization-derived friction microscopy and lithography, including single-unit-cell wear of ferroelectrics with atomic terrace edge features. Finally, we demonstrate this approach as a top-down, chemical-free and resist/maskless lithography technique, which can be potentially applied to the fabrication of three-dimensional (3D) and monolithic nanostructures when multi-pass switching and milling of the ferroelectric surface is implemented.

## Results

### Observation and investigation of tribological asymmetry in ferroelectric LiNbO$_3$ crystal

The experiments were performed on commercially available periodically poled LiNbO$_3$ (PPLN) single crystals, composed of alternating out-of-plane polarization domains, further described in Supplementary Fig. 1. We observed polarization-dependent friction and wear of the sample surface using single-crystalline conductive diamond probes (NM-TC, Adama Innovations), selected for their extreme hardness and stiffness, with relatively high contact loading force (5 μN) and scan rate (4.88 Hz, equivalently 146.48 μm/s). The down domains oriented into the sample plane are less heavily worn than the up domains oriented out of the sample plane, resulting in strongly asymmetric milling after multiple scans, as shown schematically in Fig. 1a.

Accompanying this wear asymmetry, even though the pristine surface presents a flat morphology (Fig. 1b), we observed a polarization-dependent friction contrast (Fig. 1c) during the milling scans, with higher friction in the up-oriented domains. After the 50th milling scan by the diamond probe, the surface clearly shows the height difference between up and down domains (Fig. 1d); however, as demonstrated by subsequent piezoresponse force microscopy (PFM) imaging (Fig. 1e) using the same probe, this asymmetric milling process does not affect the domain polarity, which remains stable throughout the millings. The 3D surface plot in Fig. 1f also shows a clearly visible height difference after 50 milling scans, as does the line

profile across the domains during the last milling scan in Fig. 1g, demonstrating strong height and friction differences between up and down domains. The wear depth compared with the pristine background region is approximately 4.56 nm for up domain and 3.64 nm for down domain regions (Supplementary Fig. 3), and the roughness of the surface decreases after the milling scans (Supplementary Fig. 4). In addition, cross-validation of wear asymmetry using scanning electron microscopy (SEM) further indicates the contrast arises from the height difference of patterned domains, not from the polarization difference (Supplementary Fig. 6).

Although the height and friction difference signals (Fig. 1h) oscillate because of the contact geometry difference between frame-up (scan from bottom to top) and frame-down (scan from top to bottom) during milling scans, we note that height (down − up) and friction (up − down) differences are always positive. Further, the slope of height difference (relative wear rate) is clearly correlated with the friction difference, and the trend shows four distinct regimes in Fig. 1h. Interestingly, the boundaries between each regime correspond to where we performed intermediate PFM scans (indicated by * and shaded color). In particular, to check the domain stability after the 10th, 20th and 30th milling scans, we conducted PFM scans during which we applied approximately one-tenth of the loading force, as compared to that during milling scans. After each of these intermediate PFM scans, height and friction difference trends significantly change. These findings suggest that the tribological interactions between the probe and the ferroelectric are highly sensitive and can be controlled by manipulating the probe-sample interface.

Crucially, we find that the polarization-dependent asymmetric mechanical wear varies significantly as a function of the loading force (Supplementary Figs. 7–9). As detailed in Supplementary Fig. 7, below a threshold loading force (less than 5 μN), no significant milling is observed, whereas when the force is too high (20 μN), surface deterioration and material fracture rather than steady asymmetric milling can dominate the resulting topography[22,23]. At the optimum loading force range (5–10 μN), we observed asymmetric wear of domains with some deterioration of the friction contrast and wear rate with continuous milling because of wear debris attachment of the tip, as further discussed in Supplementary Figs. 11 and 12. However, increasing or decreasing the scan rate during milling has insignificant effects on the tribological asymmetry (Supplementary Fig. 13).

Beyond micro/nanoscale wear of ferroelectric single crystals using an AFM probe, we also demonstrate scalability of this process to technologically relevant large-scale patterning by simply polishing the whole crystal using silica nanoparticles that effectively act as millions of mobile AFM tips (Fig. 1i). Figure 1j shows the digital photograph of the crystal after such polishing and the optical microscope (Fig. 1k) and SEM (Fig. 1l) images exhibit the periodic boundaries of asymmetrically patterned surfaces at large scale. As long as the nanoparticles were comparable in size to a typical AFM probe, giving rise to similar strain gradients under applied force, we could clearly see that up domains were preferably polished, resulting in nanoscale trench structures over a 9-millimeter square area of the sample as seen in the height and PFM phase (Fig. 1m, n), which is consistent with wear results observed using diamond probes. We also demonstrate this large-scale asymmetric patterning in relaxor ferroelectric PZN-PT crystals where the ferroelectric domains are randomly oriented with a lateral size of a few tens of nanometers (Supplementary Fig. 15). We believe that combining a bulk poling process with nanoscale silica bead polishing will enable us to realize bulk scale chemical-free/maskless lithography.

### The origin of tribological asymmetry

To understand the fundamental microscopic mechanisms behind our observations, we need to consider the possible interplay between friction—the resistance to relative motion between two surfaces—and mechanical wear—the removal of surface atoms after rubbing two

surfaces—which can be strongly correlated if the loading force is sufficiently high so that the probe indents the crystal during scanning. Friction and wear are complex tribological phenomena in which contributions of several possible micro/nanoscopic mechanisms could lead to the observed asymmetry of these responses in ferroelectrics[24]. Previous studies have already reported on the asymmetric lateral force microscopy signals of ferroelectric single crystals[25,26], but at a few tens of nN loading force, such asymmetry might come from the effects of screening charges or adsorbates on the surface. However, if the friction properties are indeed governed by screening conditions or adsorbates (especially chemisorbed species), the asymmetry should disappear with continuous milling scans, as any surface species or asymmetric skin layers are gradually removed. This is not the case in our results. Rather, we find that the asymmetry persists throughout the full cycle of continuous milling.

Beyond the surface electrochemistry mechanism described above, another possibility is that the asymmetry emerges due to the different mechanical properties of up and down domains induced by the flexoelectric field generated by the non-uniform strain applied when the AFM tip is pressed onto the sample surface. Although the ferroelectric remanent polarization is itself symmetric, this additional flexoelectrically driven contribution either competes with or enhances it and thus leads to a different effective polarization in the up and down domains. This strain-gradient-induced polarization has been shown to produce asymmetric mechanical responses[12,27,28]. At the same

loading force in up and down domains, we should therefore expect a difference in applied stress at the contact area during the milling scan, as schematically illustrated in Fig. 2a. Because applied stress at a larger contact area region could induce higher friction, and thus more wear[29], we consequently expect higher friction and wear in up domains than down domains (Fig. 2b, c). Specifically, in ferroelectric up domains, larger stress at the junction of tip and sample can be applied, or a deeper ploughing depth of the tip into the sample can be expected during the scan.

Alternatively, in the second possible mechanism shown schematically in Fig. 2d, tribological asymmetry can arise electrostatically if the loading force is sufficiently high (around a few micronewtons) to enable screening charges to be scraped off the ferroelectric surface[22,30]. The unscreened ferroelectric surface of up vs. down domains has opposite electrostatic potential and field, which would lead to an asymmetric electrostatic force between the tip and sample[31]. Furthermore, related to the triboelectric effect, charge transfer during sliding of the tip over the opposite polarization surface can induce asymmetric friction because of the potential difference[32–34].

We first examined the role of flexoelectricity in the friction responses using a self-consistent approach that couples flexoelectricity and piezoelectricity[35] based on a linear continuum theory of piezoelectricity[36]. To confirm that flexoelectricity can explain the observed tribological asymmetry between ferroelectric up and down domains, we performed finite element simulations to quantify the

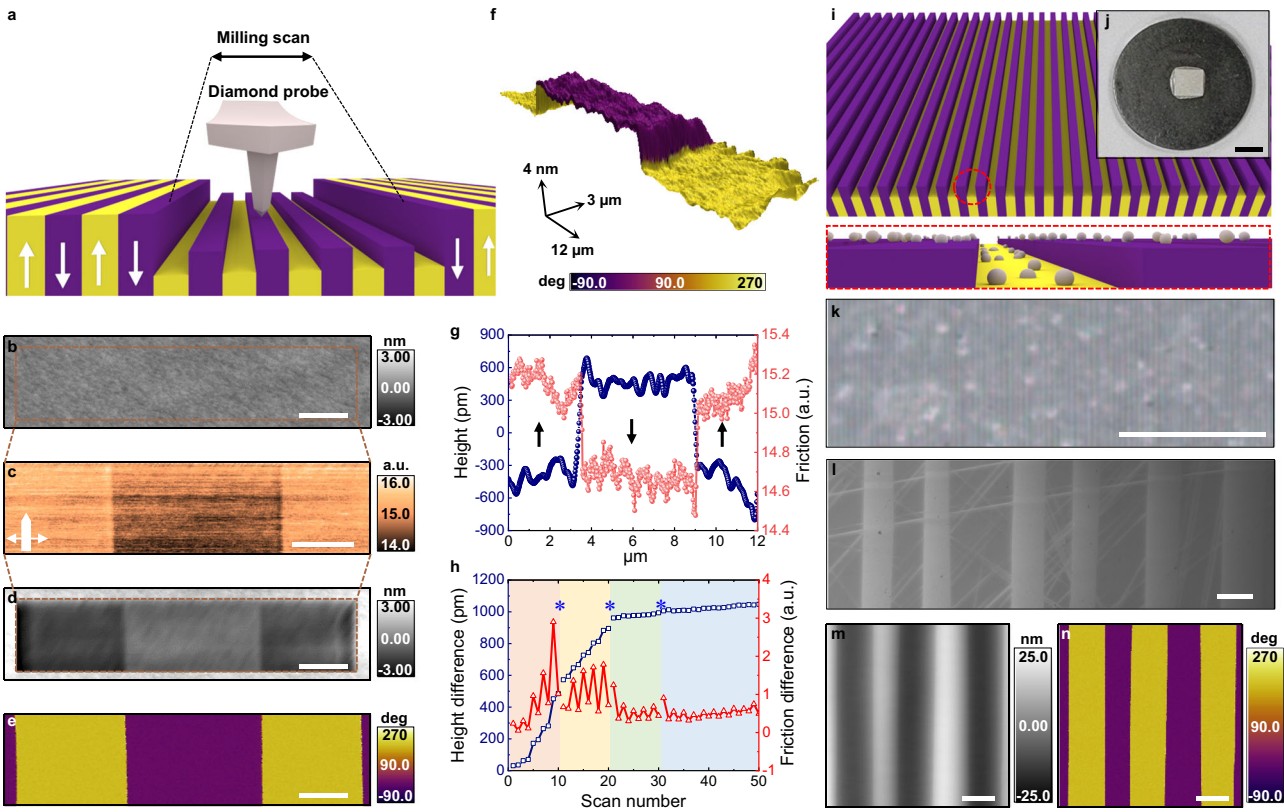

**Fig. 1 | Observation of asymmetric friction and wear of ferroelectric LiNbO₃ single crystal. a** Schematic of asymmetric milling of periodically poled lithium niobate (PPLN) using a diamond probe. Ferroelectric up and down domains exhibit different mechanical wear rates after milling without any external voltage applied to the probe. Pristine up and down domains begin with identical initial topography but show different heights after repeated milling scans. **b** AFM height before milling, **c** friction image acquired during the 50th milling scan (scan angle of 90° with the fast scan axis perpendicular to domain walls). **d**, **e** AFM height (**d**) and PFM phase (**e**) after fifty milling scans show stable domain orientation after the wear process. Scale bars in (**b**–**e**) are 2 μm. **f** 3D surface plot of the patterned surface after

50 milling scans with a color scale indicating the PFM phase. The full-length scale along each respective axis is shown with an arrow. **g** Height and friction signal during the 50th milling scan. Arrows indicate the polarization orientation. **h** Height and friction difference between up and down domains with increasing scan number. After 10th, 20th and 30th milling scans (indicated by * and shaded color), intermediate PFM scans were conducted. **i** Schematic of large area milling by simply polishing the crystal using silica particles. **j**–**l** Digital photograph of 3 × 3 mm² PPLN (**j**), Optical microscope image (**k**) and scanning electron microscope image of patterned PPLN (**l**). **m**, **n** AFM height (**m**) and PFM phase (**n**) of large area milled PPLN. Scale bars are 3 mm for (**j**) 300 μm for (**k**) and 5 μm for (**l**–**n**).

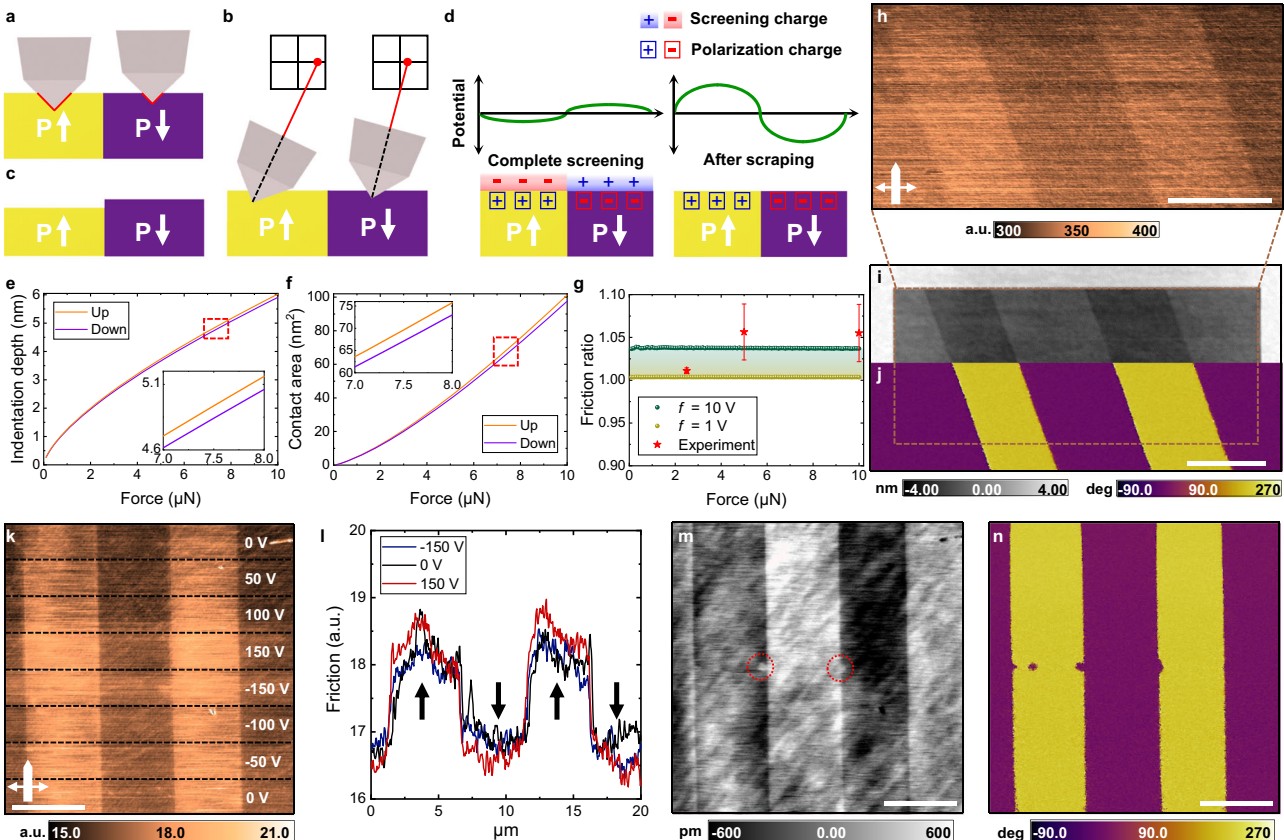

**Fig. 2 | Origin of asymmetric friction and wear of ferroelectrics. a–c** Schematic of flexoelectrically induced mechanism, in which the asymmetric tribological properties depend on the direction of the out-of-plane polarization, leading to a larger contact area during the scanning in up domains. The probe sinks deeper in the ferroelectric up domain than the down domain because of stiffness asymmetry (**a**). During the dynamic scan, the absolute value of the lateral signal in the position-sensitive photodiode is higher in the up domain than the down domain because of the larger contact area and indentation depth (**b**). The resulting height after continuous scans is higher in the down domain due to continuous milling with higher friction in the up domain (**c**). **d** Schematic of unscreened surface charge-driven mechanism. **e–g** Computational mechanics approach of asymmetric friction in LiNbO₃. Indentation depth (**e**) and contact area (**f**) of up and down domains based on cubic flexoelectricity with equal longitudinal and transversal coefficients corresponding to a flexocoupling coefficient of 10 V. Insets are magnifications of the

red dotted area. Friction ratio (up/down) obtained from the numerical simulation and experiments (**g**). Flexocoupling coefficients simulated in (**g**) are between 1 and 10 V. Ten measurement points were averaged for each data point with error bars given by standard deviations. **h–j** Investigation of tribological asymmetry with a non-conductive diamond probe. Friction during the milling scan with non-conductive probe (**h**), resulting AFM height (milled inside) with pristine background region (**i**) and PFM phase (**j**) using conductive diamond probe after one milling scan. **k–n** Results of high voltage application to tip during the milling. Friction during the 1st milling scan (**k**) and line profiles in the 150 V, 0 V and −150 V regions (**l**). Arrows indicate the polarization orientation. Resulting AFM topography (**m**) and PFM phase (**n**) after 10 milling scans (scan angle of 90° with the fast scan axis perpendicular to the domain walls). Red circles indicate higher topography in electrically switched regions. Scale bars in (**h–n**) are 5 μm.

indentation depth and contact area depending on polarization direction in the presence or absence of flexoelectric coupling, using the Signorini-Hertz-Moreau model for contact[37]. The AFM tip is idealized as a spherical rigid indenter in contact with an ideally flat LiNbO₃ surface. Further details of the theoretical and computational model are described in Supplementary Information.

Indentation with the spherical diamond probe generates a spreading electric potential distribution in the LiNbO₃ crystal (Supplementary Fig. 18). Since flexoelectric polarization always results from downward pressure-induced strain gradients and thus interacts differently with up and down-oriented ferroelectric domains, this electric potential distribution varies depending on the direction of out-of-plane polarization. Piezoelectric and flexoelectric polarization fields upon indentation (Supplementary Fig. 19) thus lead to different competing or synergistic combinations in up and down domains. This combined interaction results in systematically higher indentation depths and contact areas for up domains (Fig. 2e, f), and the asymmetry disappears in the absence of flexoelectricity (Supplementary Fig. 20). Furthermore, the differences in the indentation depths and contact areas increase as flexoelectric coupling becomes stronger

(Supplementary Fig. 21). Presuming that the contact area and friction have a linear relation as predicted by single asperity friction models[38], the ratio of contact areas between up and down domains should be equal to the ratio of their friction, which is an experimental observable. From the simulation, this ratio is found to be independent of the indentation force (Fig. 2g). Experimental values of the friction ratio in Fig. 2g, acquired with changing loading force (Data from Supplementary Fig. 7), are consistently in agreement with simulated values for a flexocoupling coefficient of 10 V[5,39]. Experimentally, at lower loading forces (e.g., 2.5 μN), contact conditions deviate from ideal indentation during scanning, and thus, no agreement with the simulated results should be expected.

To further discriminate between the flexoelectric vs. electrostatic mechanisms, we conducted nanotribology experiments using a non-conductive single-crystalline diamond probe (D300, *SCD Probes*) to test the strength of possible electrostatic contributions. The probe was demonstrated to be non-conductive from the *I−V* curve measurement shown in Supplementary Fig. 23 but was still equivalently hard and stiff for mechanical wear of ferroelectric PPLN. Despite the non-conductive nature of the probe, we observed the same asymmetric behavior, with

up domains having higher friction than down domains, as seen in Fig. 2h, even with one milling scan at a loading force of 6 μN and scan rate of 4.88 Hz. Height and PFM phase measured using a conductive probe after mechanical scraping (Fig. 2i, j) still remain stable, confirming that the asymmetry originated from pure mechanical milling of the probe on the ferroelectric surface. This finding can explain why electrically disconnected and insulating silica beads can mechanically pattern the PPLN surface over a large area exactly the same way as the conducting diamond probe does over a small area.

Moreover, to further investigate the role of surface electrostatics from the other extreme, we tested electrostatic effects on the friction and wear of PPLN when applying an electric field. If the observed asymmetric friction and wear arise from the electrostatic asymmetry of the surface, we would expect high voltage application during milling to significantly change the results. However, when we applied such high voltage, varying from −150 V to 150 V to the tip during milling scans, we observed no notable change in either the magnitude of the friction signal or the contrast between the up and down domains during milling (Fig. 2k, l). Furthermore, the resulting topography following 10 scans performed with the same varying applied bias shows no significant height change, as can be seen in Fig. 2m. We do, however, observe ferroelectric switching from up to down domain orientation occurring just as the voltage was varied between −150 V and 150 V, as shown in Fig. 2n. We note that these switched regions are slightly higher in topography than the un-switched up domain (Fig. 2m, n), further indicating the switchable nature of the wear asymmetry. We therefore exclude the electrostatic effects as a dominant contribution to the observed asymmetric nanotribological response.

In addition, we note that highly sensitive wear behavior depending on loading force (Supplementary Figs. 7–9) and tip condition (Supplementary Figs. 11–13, 24, Supplementary Table 1) supports the mechanism for tribological asymmetry governed by mechanical stress applied to the surface. Consequently, the observed tribological asymmetry is comparable to previously reported layer-by-layer wear[40] but is distinct from atom-by-atom wear[41], as thermal activation is not dominant (Supplementary Fig. 25).

Taken together, our simulation and experimental observations, including other possibilities described in Supplementary Information, suggest that the dominant mechanism at the origin of the asymmetrical tribology is flexoelectrically induced polarization and the consequent asymmetry of mechanical responses between oppositely polarized ferroelectric up and down domains. We note that this asymmetry implies an anomalous, positive correlation between the hardness of the domains and their wear rate, while normally high-hardness materials are more resistant to wear, as reported in the form of an Ashby plot between hardness and wear coefficient[42]. Here, the flexoelectrically modulated mechanical properties mean that up domains have larger hardness[12] yet are more heavily worn. To elucidate the anomaly, nanoindentation was performed on ferroelectric up and down crystals to determine the hardness and penetration depth difference between the up and down domains. Interestingly, we found that the indentation depth measured in a low strain gradient regime differs from that in a high strain gradient regime. More details of this observation can be found in Supplementary Fig. 27. Furthermore, the tribological asymmetry is indeed switchable, where the electrically reversed down domains exhibit lower friction signals than the up domains (Supplementary Fig. 28).

## Polarization-derived lithography using switchable tribology of ferroelectrics

Using our observation of asymmetric wear, we could fabricate diverse nanostructures on the ferroelectric single crystal, manipulating the domain configuration by applying electrical bias using a nanoscale probe, as shown in Fig. 3a. This also enables 3D tomographic studies of ferroelectric nanostructures with continuous milling scans to map the

full evolution of the pillar structure during subsequent milling, as schematically depicted in Fig. 3a. To demonstrate, in the up domain of the initial PPLN configuration, we switched a 4 × 4 array of nanoscale circular down domains by applying 150 V to the stationary AFM tip on the ferroelectric surface at the designated array positions for 120 s. Since down domains are more resistant to mechanical wear than up domains, this domain pattern allowed us to create an array of nanopillars milled using a diamond probe. The corresponding height, PFM phase and PFM amplitude are detailed in Supplementary Fig. 29, while the height and PFM phase after ten milling scans with 5 μN at 1.95 Hz are shown in Fig. 3b, c. The pillar height saturates at approximately 14.13 nm around the 60th milling scan (Fig. 3d). We note that initially, and for a small number of milling scans, the wear process yielded well-defined low nanopillars with a cylinder-like structure and a flat top. However, as the milling continues, progressive wear of the down-oriented domains also leads to a more cone-like morphology, with a rounded top as the full growth of the pillars can be seen in Fig. 3e.

The same lithographical concept can be applied to thin film ferroelectrics where more precise friction visualization and lithography techniques are possible, although the procedures may require greater care because decreased film thickness promotes stronger strain gradients and can lead to flexoelectric switching[43]. We demonstrated that this polarization-derived lithography could be successfully reproduced at reduced crystal dimensions, such as in a 100 nm LiNbO$_3$ thin film. Simply switching ferroelectric domains electrically and milling at an optimum loading force regime can create a desired nanostructure (Supplementary Fig. 30). Height, PFM phase and 3D surface images with color overlapped with their PFM phase in Fig. 3f–h exhibit that facile fabrication without any chemicals or photomask is realized after artificially decorating the thin film with the text "FERRO" with down domains, and "LITHO" with up domains before milling. Height and PFM phase line profiles (Fig. 3i) clearly show the down domains as higher than up domains in the patterned region. To demonstrate the generality of this phenomenon and its independence of chemical composition and crystal structure, in addition to uniaxial LiNbO$_3$ single crystal and thin film, we carried out polarization-derived lithography in tetragonal PbTiO$_3$ thin film ferroelectrics. Once again, we observe that locally switched down domains (Fig. 3j) have the lower friction signal (Fig. 3k). As shown in Fig. 3l, atomic PbTiO$_3$ terrace edges, which originate from the SrTiO$_3$ substrate (Supplementary Figs. 31 and 32), were revealed after multiple milling scans at gradually increased loading force (up to 1600 nN). Surprisingly, the height difference between patterned up and down domains is around 4 Å, which implies the height difference is indeed one unit cell height calculated from the XRD data (4.144 Å).

Furthermore, reproducible wear rate control based on a choice of polarization orientation, which can also be configured during the milling process, allows for the fabrication of a more complex 3D nanostructure. Figure 4a shows a schematic of such a multistep lithographic process at different stages of fabrication. From the pristine domain state (up in this case), in which the desired domain nanostructure can be patterned by applying an electric bias to the scanning probe, initial mechanical milling establishes the desired height difference between the down and up domains. Subsequent rewriting of the domain structure and additional mechanical milling can then be repeatedly alternated to create the target nanostructure with multiple height levels. A final switching step can then be carried out to obtain a uniform polarization orientation throughout the sample.

A proof-of-concept example of this approach to develop complex 3D structures is shown in Fig. 4b–f. The complete evolution of the 3D structure during repeated writing and milling is shown in Supplementary Figs. 33 and 34. The down-oriented rectangular domain and the text "FERRO LITHO" are first electrically patterned and milled before the up-oriented schematic AFM probe is electrically patterned.

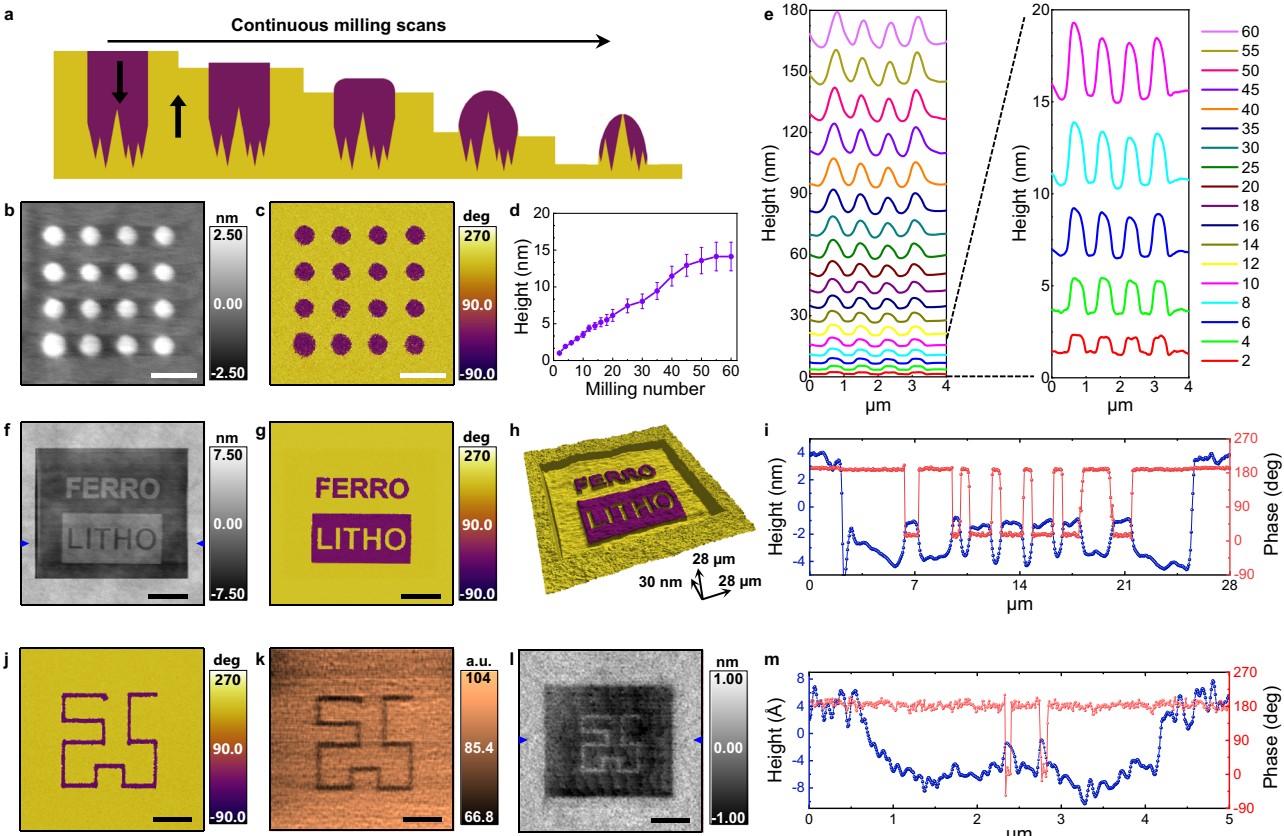

**Fig. 3 | Universal tribological asymmetry across a broad materials space.** The tribological asymmetry works universally regardless of the size and type of ferroelectric. **a**–**e** Nanopillar fabrication on LiNbO₃ single crystal by continuous mechanical milling after electrical switching, both with the same conductive diamond probe. Schematic of nanopillar fabrication (**a**). AFM height (**b**) and PFM phase (**c**) after ten milling scans at a loading force of 5 µN and a scan rate of 1.95 Hz. Scale bars in (**b**) and (**c**) are 1 µm. Average pillar height evolution with continuous milling scans (**d**). Pillar height was averaged from the line profiles across four pillars with 20 data points. The error bars indicate the standard deviations. Growth of nanopillars during continuous milling scans based on the line profiles from four pillars (**e**). Height data were acquired after the milling scan from 2 to 60 scans, as indicated by the color key. **f**–**h** Ferroelectric nanostructure fabrication on thin LiNbO₃ film. AFM height (**f**), PFM phase (**g**) and 3D surface images with color overlapped with their PFM phase (**h**). Scale bars in (**f**) and (**g**) are 6 µm. The full-length scale along each respective axis is shown with an arrow for (**h**). **i** Height and PFM phase line profiles along the blue marker in (**f**). **j**–**m** Nanofabrication of PbTiO₃ thin film. PFM phase after the artificial decoration of ferroelectric domains (**j**), Friction image during milling scans at 800 nN (**k**) and AFM height after multiple milling scans (**l**). Scale bars are 600 nm for (**j**) and (**k**) and 1 µm for (**l**). **m** Height and PFM phase line profile along the blue marker in (**l**).

The AFM tip feature is fabricated by additional milling scans after this switching step. The height and PFM phase (Fig. 4b, c) show the 3D ferroelectric nanostructure after the 2nd milling process. The height image in Fig. 4d, including both pristine and milled areas, indicates the different topography, as do the line profiles in Fig. 4e in which the distinct 3D structure can be discerned. We note that the use of a nanoscale probe allows us to create complex structures of nanoscale ferroelectric domains, and therefore, the resulting mechanical lithography shows significant technological promise. Although many previous studies on scanning probe lithography successfully carried out sample structuring, with a height difference obtained via mechanical, thermal or chemical patterning[17,18,44], the present work uniquely demonstrates scanning probe nanostructuring using the tribological asymmetry between domains with different polarization orientations.

## Discussion

To summarize, our work reveals the asymmetric friction and wear of ferroelectrics, which opens up an alternative way toward probing and manipulating ferroelectric domains based on the switchable tuning of their tribological properties. We determine that the higher friction and thus faster wear rate in up domains originate from the strain-gradient driven flexoelectric response, which either competes with or enhances the ferroelectric polarization in oppositely oriented domains, resulting in higher friction in up domains than down domains. Furthermore, our findings enable us to propose a simple methodology for patterning a desired 3D structure with arbitrary complexity by alternating electrical switching and mechanical milling steps. Finally, we establish the universal nature of tribological asymmetry independently of chemical composition or crystal structure and demonstrate single-lattice scale wear in epitaxially grown ferroelectric thin films. We envision that this top-down, chemical/resist-free and maskless lithography technique can be scalably applied in the fabrication of ferroelectric nano/ microstructures.

## Methods

### Materials

Three different LiNbO₃ samples were used to probe the asymmetric friction and wear. A periodically poled lithium niobate (PPLN) single crystal (AR-PPLN test sample, 3 × 3 mm² with a thickness of 500 µm, Asylum Research, Oxford Instruments) was chosen for the primary experiments demonstrating the polarization-dependent asymmetric tribology from Figs. 1, 2, 3a–e, Supplementary Figs. 1–14, 24, 25, 28 and 29. A stoichiometric LiNbO₃ (optical grade single crystal, 10 × 10 mm², with a thickness of 500 µm, MTI Corporation) was used in control experiments to investigate the possible effects of variations in defect density and nanoindentation measurements (Supplementary Figs. 26 and 27). For prototype nanostructure fabrication (Figs. 3f–i and 4 and Supplementary Figs. 30 and 33–35), 100 nm thick z-cut undoped

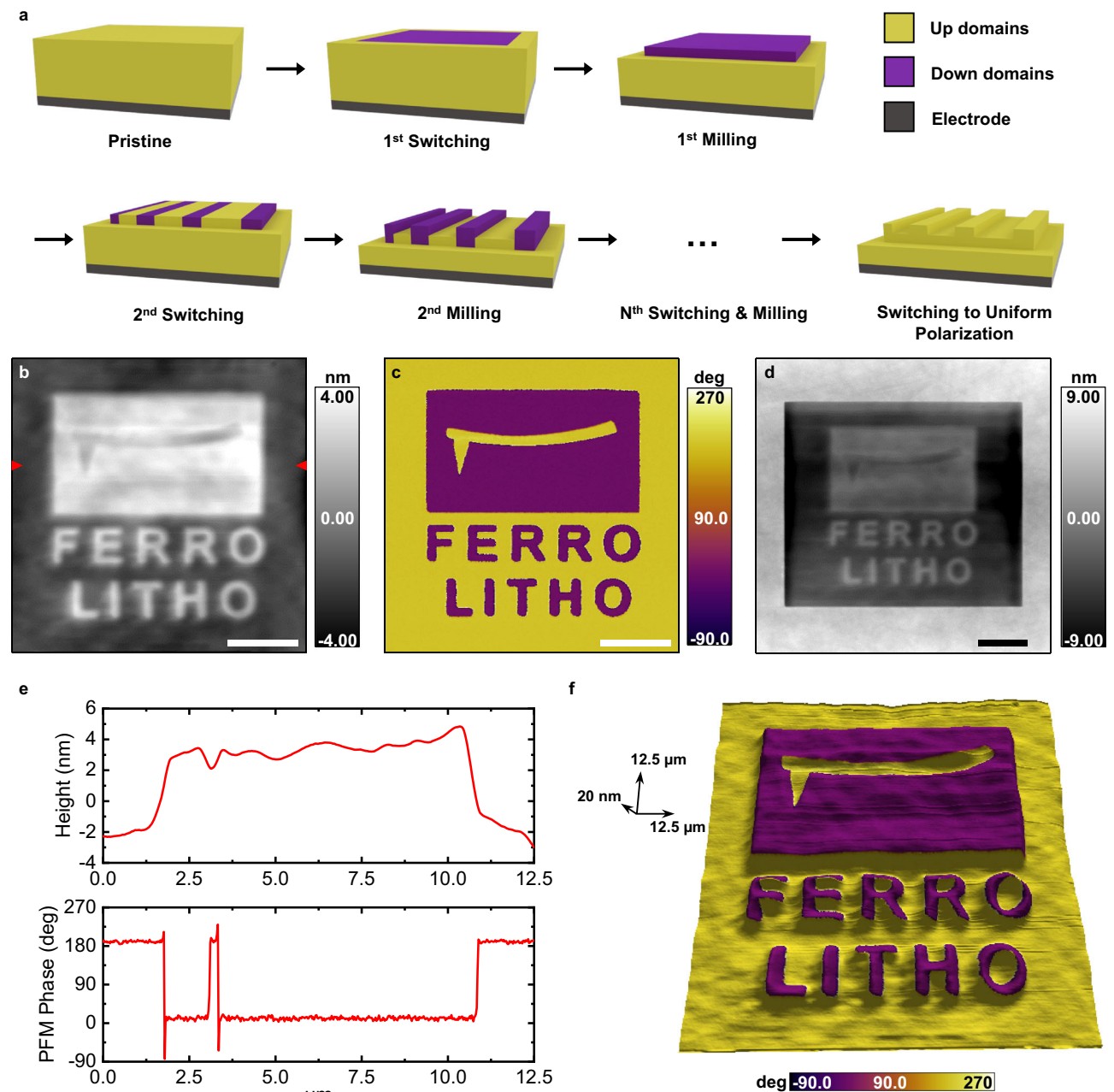

**Fig. 4 | 3D nanostructure fabrication using asymmetric nanotribology on ferroelectric LiNbO₃ thin film. a** Schematic of top-down, chemical-free and maskless multi-patterning combining the switchable nature and asymmetric wear of ferroelectrics. **b**, **c** AFM height (**b**) and PFM phase (**c**) after multiple switching and patterning steps. **d** AFM height, including pristine background area, after multiple patterning. **e** Line profiles of height and PFM phase along the AFM probe feature in (**b**). **f** 3D representation of the multi-patterned structure with PFM phase color superimposed on the height image. Scale bars are 3 µm for (**b**–**d**). The full-length scale along each respective axis is shown with an arrow for (**f**).

LiNbO₃ films on Cr/SiO₂/LiNbO₃ substrates with z+ oriented polarization (NanoLN) were prepared by cutting from single crystal congruent LiNbO₃ using the ion-slicing method[45].

The PbTiO₃ thin film in this study was grown epitaxially using off-axis radio frequency magnetron sputtering on a TiO₂-terminated, (001)-oriented SrTiO₃ substrate (CrysTec GmbH). A SrRuO₃ layer was first deposited at 640 °C in 100 mTorr of O₂/Ar mixture of ratio 3:60 using a power of 80 W with a stoichiometric target. The PbTiO₃ film was then grown in situ in 180 mTorr of a 20:29 O₂/Ar mixture using a power of 60 W at a temperature of 540 °C and a Pb₁.₁TiO₃ target with a 10% excess of Pb to compensate for Pb volatility. The SrRuO₃ layer is 60 ± 5 unit cells thick, with a c-axis = 3.975 ± 0.002 Å. The PbTiO₃ layer is 210 ± 8 unit cells thick, with c-axis = 4.144 ± 0.001 Å. The as-grown sample is atomically flat, with a roughness of the order of 2 Å (Supplementary Fig. 31).

PZN-5.5%PT (001) single crystals (5 × 5 mm², a thickness of 500 µm, Microfine) were prepared to demonstrate large-area asymmetric patterning using silica nanobeads (Supplementary Fig. 15).

## Scanning probe microscopy and polarization-derived lithography

All scanning probe measurements were performed using a commercial atomic force microscope (Cypher ES and Cypher VRS, Asylum Research, Oxford Instruments). Single-crystalline conductive diamond tips (NM-TC, Adama Innovations, Lot number: 009-013) were used to image the surface friction response and to grind the LiNbO₃ samples.

Ferroelectric LiNbO$_3$ nanopillars were fabricated using a single-crystalline diamond probe (NM-TC, Adama Innovations, Lot number: 009-013), where electrical switching and mechanical milling were conducted using the same diamond probe. Au-coated Si probes (4XC-GG, MikroMasch) were used for visualization and lithography of PbTiO$_3$ thin film to minimize the effect of flexoelectric switching. After the milling scans of PbTiO$_3$ thin films, a pristine probe was used for higher resolution. During the high-force contact imaging, DC or AC electric bias was not applied through the probe except during the simultaneous piezoresponse force microscopy (PFM) imaging at high contact force in Supplementary Fig. 10. The loading force during the scan was calculated by multiplying the inverse optical lever sensitivity obtained from the force-distance curve measurements, the spring constant of the probe, and set point difference in the position-sensitive photodiode in contact with the sample. The scan angle was fixed at 90° (perpendicular to the long axis of the cantilever) during friction imaging and milling.

PFM was performed to image polarization domains at a much lower loading force compared to the milling scans to prevent sample damage during imaging, except in the case of checking for the possible transient switching during high force application, as shown in Supplementary Fig. 10. During some of the measurements, temperature and humidity were maintained at set levels in the environmental cell with an in-house-designed low-noise humidity controller to ensure stable condition of the ferroelectric surface[46,47]. During the switching process in the nanostructuring experiments in LiNbO$_3$ thin film, Pt-coated Si probes (HQ:DPER-XSC11, MikroMasch) were used to switch the rectangular and "FERRO LITHO" domain from up to down in the first switching to prevent tip contamination before milling scans. After this first switching, milling scans were conducted under optimum conditions using diamond probes. The same diamond probe was used in the second switching, creating an AFM probe feature in the rectangular down domain, following multiple milling scans and switching to obtain a uniform polarization state.

### Large-scale patterning of ferroelectric single crystals

PPLN single crystals were gently milled and polished using the Multiprep™ Polishing System (Allied). Following mechanical grinding of the crystal using diamond lapping films (1, 3, 9, and 15 μm, Allied), final polishing was performed with colloidal silica suspension (40 nm, Allied) on a synthetic polishing cloth (Vel-Cloth, Allied). During polishing, the platen speed was 150 rpm at 2.94 N (300 gF) for 3 min. As control experiments, three different samples were prepared. We conducted mechanical grinding of the crystal using diamond lapping films, dipping in the colloidal silica solution and polishing using colloidal suspension, separately in three pristine PPLN (Supplementary Fig. 14). We further demonstrated asymmetric nanopatterning in relaxor ferroelectric PZN-5.5%PT single crystals where ferroelectric domains have different size and distribution. As such, technically feasible chemical and mask-free large-area patterning can be achieved through a simple polishing process of ferroelectric materials (Supplementary Fig. 15).

### Flexoelectrically coupled contact mechanics simulations

The complete model of flexoelectrically coupled contact mechanics is described in the computational model and contact model sections in the supplementary materials. We used the model to compute the indentation depth and contact radius of a spherical diamond tip indenting a poled LiNbO$_3$ sample, explicitly accounting for flexoelectricity. A rectangular computational domain is considered, with $L = 40$ nm and $H = 20$ nm. The material parameters for elasticity, dielectricity and piezoelectricity are taken from an open database of computed material properties[48]. We considered a simplified form of the strain gradient elasticity tensor $h_{ijklmn} = (\lambda\delta_{ij}\delta_{lm} + 2\mu\delta_{il}\delta_{jm})\ell^2\delta_{kn}$, where $\lambda$ and $\mu$ are Lamé elasticity parameters, $\ell$ is the length-scale taken as 10 nm, and $\delta_{kn}$ is the Kronecker delta. We assumed a cubic isotropic

form of the flexoelectric tensor[49], with two independent components, longitudinal $\mu_L$ and transversal $\mu_T$, the shear component being $\mu_S = \frac{1}{2}(\mu_L - \mu_T)$. In all simulations in this work, we took $\mu_L = \mu_T = \kappa f$, where κ is the dielectric tensor and $f$ is the flexocoupling coefficient; thus, $\mu_S = 0$. The 3D solution of the axisymmetric problem is shown in Supplementary Fig. 18 considering an upward polarized sample, for $f = 10$ V. The boundary conditions of the corresponding axisymmetric problem are as follows: left side clamped horizontally and bottom side grounded and clamped vertically.

### Estimation of friction force

To quantitatively compare the simulated asymmetry in the indentation depth to that observed experimentally in the friction, we assumed a linear relation between the friction force and the contact area[38,50,51]. Consequently, the ratio between contact areas in up and down domains should be equal to that of their friction. For a spherical indenter, we can write $F_f = \tau A_c = \tau 4\pi R^2 \left(1 - \cos\sin^{-1}\left(\frac{R_c}{R}\right)\right)^2$ where $F_f$ is the friction force, $\tau$ is the shear strength, $A_c$ is the contact area, $R_c$ is the contact radius and $R$ is the radius of the sphere (10 nm). The ratio $\frac{F_f^{up}}{F_f^{down}} = \frac{A_c^{up}}{A_c^{down}}$ is given in Supplementary Table 2 for different values of flexocoupling coefficient.

### Reporting summary

Further information on research design is available in the Nature Portfolio Reporting Summary linked to this article.

### Data availability

All data used in the main text are available on Yareta (https://doi.org/10.26037/yareta:yqifa2kr3valpkbzzk7g6tvqlu, reference[52] or Github (https://github.com/MIIMSEKAIST/Switchable-tribology-of-ferroelectrics-Data-sets)). Other datasets in Supplementary Information are available from corresponding authors upon request.

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

## Acknowledgements

S.H. acknowledges the National Research Foundation of Korea (NRF) grant funded by the Korean government (MSIT) (Grant No. 2020R1A2C2012078, NRF-2022K1A4A7A04095892 and RS-2023-00247245) and the KAIST-funded Global Singularity Research Program for 2022 and 2023. P.P. acknowledges Division II of the Swiss National Science Foundation under project 200021_178782. S.C. acknowledges the National Research Foundation of Korea (NRF) grant funded by the Korean government (NRF-2018-Global Ph.D. Fellowship Program). I.A. acknowledges the support of the European Research Council (No. StG-679451 to I.A.), the Spanish Ministry of Economy and Competitiveness (No. RTI2018-101662-B-I00), and the Generalitat de Catalunya (ICREA Academia award for excellence in research to I.A. and Grant No. 2017-SGR-1278 to I.A.). CIMNE is a Severo Ochoa Centre of Excellence (2019-2023) under the grant CEX2018-000797-S to I.A. funded by MCIN/AEI/ 10.13039/501100011033.

## Author contributions

S.C., P.P. and S.H. conceived the idea of asymmetric milling. S.C., I.G. and K.C.-E. designed the experiments. S.C. performed milling and scanning probe measurements. J.B.-M. and I.A. conceived the computational study and analyzed the results. C.L. prepared the PTO thin film sample. S.C., P.P. and S.H. analyzed the SPM data. J.Y., L.M. and G.C. supported the data analysis. D.K. and S.M.H. performed nanoindentation measurements. S.C. and H.K. performed SEM measurements. All authors discussed the results and edited the manuscript.

## Competing interests

The authors declare no competing interests.
