## [Peer Review File · Nature Communications]

REVIEWER COMMENTS

Reviewer #1 (Remarks to the Author):

This paper has a large amount of excellent science. Unfortunately it is also a very hard read, and will confuse most people, reducing its impact. The most obvious problem is that it has 23 pages for the main text, and 53 for the Supplemental, with most of the hard science in the SM.

In my opinion this work should be split into two papers which are related, but are fundamentally different:

Paper 1: Analysis of the coupling between FeX and ferroelectricity. This would include the theory and parts of the experiments. The authors would need to demonstrate reasonable agreement between the theory and experiments, which is not completely clear.

Paper 2: The work involving wear to produce different shapes etc.

By splitting the paper it will become much, much more readable. In addition, the authors can provide much more useful information to the larger community. While the two papers could, perhaps appear in the same journal, there are reasons to consider the wear paper appearing in a more tribology related journal. Even though the science is very high quality, I cannot recommend the paper for further processing in its current form.

Some additional comments, in no particular order:

Are there differences in fracture toughness depending upon the polling direction? This matters for wear.

1. Please don't use the term "etching" anywhere, that has an accepted usage with solvents.
2. An important experiment would be to use tips of different radii to prove that FeX matters. The authors could also analyze results for different silica, rather than just saying that comparable sizes work. (The force applied will also matter.)

3. How do their friction and wear anisotropies compare to other work in the literature?

4. What is the mechanism of what they call “etching”? Just calling it wear is not enough. Is it similar to atomistic on layer-by-layer wear, see for instance DOI 10.1080/09506608.2016.1213942. Is there ploughing?

5. On p6, line 158 the statement about adsorbates is only right if the pressure is low enough – remember 1 monolayer in 10^{-6} seconds.

6. Similarly, “scraping off screening charges” seems inappropriate, as electrons move really fast.

7. In S5 how are the values for the strain-gradient terms calculated? There is one estimate by Stengel, but I do not see any values in the paper.

8. In S5 what boundary conditions are being applied on the free surface of the ferroelectric? As written the equations may be clear to the authors, but not to the general audience. Most readers will have no idea (I don't) why a saddle-point problem is needed, and whether the “weak form” means anything.

9. How large are the coupling terms? It would be of general importance to know whether the FeX and strains can be considered as additive, or there are large coupling terms so a fully self-consistent solution is needed. From Fig. S20 they do not look large.

10. In S6 why is the nominal value from the manufacturer of the tip used? Under conditions where there is wear of the substrate one expects wear of the tip.

11. The model in S6 the authors use appears to reduce to a standard Hertz model, where the FeX contributions have already been analyzed by Mizzi et al and others more recently in the mechanics literature. Indeed, the plot they show looks just like a classical Hertzian solution. It is important to know if these results are the same or different. Currently this other work is not cited.

12. In S8 the statement that the Pt/Ir probe did not yield asymmetric wear (don't use etching please) needs explanation. This is a very interesting result.

13. In S9 the statement at the end needs to be “the presence of pre-existing defects”. Wear is due to defects, often dislocations or fracture.

14. In S10 the strain fields for different tips and hence the coupling is very different – this needs to be mentioned.

15. In S12 and elsewhere, what is the wear rate? Are we dealing with multiple layers per pass or what? Based upon Fig S2 it looks like at least a monolayer per scan.

16. What is the roughness of the surface after the wear? This matters to know if we are talking about fracture or something different.

17. What is “a.u.” in Figure S3d?

18. Why does Fig S18. Have polarization in the vacuum?

19. What is the surface charge on the diamond tip?

20. I am dubious about the relevance of Fig S21.

21. The “TiO₂” terminated SrTiO₃ substrate appears in Fig S28 without any other mention. There is also nothing about how this was prepared – I assume it is some double-layer from buffer etch+anneal, but that needs to be stated. I am not sure about the relevance of Fig. S28.

22. In Table S1 they state isotropic FeX – that is very dubious, although it may not be far off. Remember that the FeX coefficient for STO are different for [001] and [110].

Reviewer #2 (Remarks to the Author):

This manuscript makes an excellent point, artificially induced asymmetric tribological properties of ferroelectrics, that offers an alternative route to visualize and control ferroelectric domains. The switchable friction and wear behavior of ferroelectrics using a nanoscale scanning probe can be used as

smart masks, and the asymmetry is enabled by flexoelectricity coupled polarization in the up and down domains under a sufficiently high contact force. The polarization-sensitive tribological asymmetry is widely applicable across various ferroelectrics with different chemical composition and crystalline symmetry. And using this switchable tribology and multi-pass patterning with a domain-based dynamic smart mask, this manuscript demonstrated three-dimensional nanostructuring exploiting the asymmetry wear rates of up and down domains. The findings establish that ferroelectrics are electrically tunable tribological materials at the nanoscale for versatile applications. I hence recommend it for publication.

However, the authors should probably consider the following issue before producing a final manuscript:

The asymmetric tribological properties in this manuscript were demonstrated in PPLN (periodically poled lithium niobate), in which the polarization in the adjacent domains were up and downward aligned. Would the authors verify the asymmetric tribological properties in other ferroelectrics, where the polarization might be rotated and be not distributed up and downward in the adjacent domains?

Reviewer #3 (Remarks to the Author):

The manuscript by Cho et al presents a novel and impressive new method to pattern materials at small scales by exploiting a mechanism that itself is newly discovered: the dependence of the wear rate of a ferroelectric material on its local polarization orientation. In particular, regions of ferroelectrics poled “up” or “down” (with respect to the surface normal) are both worn when a diamond AFM tip is slid in contact over it under ambient conditions, but the wear occurs at different rates. This produces a height difference between regions of up vs. down polarization. The authors show the effect occurs between a minimum and maximum range of applied loads, and leads to height differences of a few nm before leveling off. A difference in friction is also observed between up- and down-polarized domains. The authors then show numerous ways the effect can be exploited for characterizing the orientation (up vs down) of previously poled ferroelectric domains; maskless patterning; and tomographic probing. They also show how polishing with a slurry of small particles can also produce the same effect, but across an entire wafer, in their words, “simply polishing the whole crystal using silica nanoparticles that effectively act as millions of mobile AFM tips.”

The results are innovative and interesting, and highly novel. The authors present multiple examples using different piezoelectric materials, demonstrating the effect is reproducible and can be well-controlled. The experimental skill required to conduct this study is substantial.

The authors present a finite element model to help explain the results. However, the main shortcoming of the paper is that the physical mechanism behind the effect is not well-explained, and other possible mechanisms (discussed below) are not considered. To be sure, this is a very complex system, as it involves: a nanoscale contact, where the stresses will be non-uniform and may even deviate from continuum predictions; the combined and interacting effects of flexoelectricity and piezoelectricity; and the very complex challenge of understanding friction and wear and their interaction, for which few

robust, physical-based theories exist even for simple materials, let alone the complex ones studied here which involve multiple coupled physical phenomena.

So, it would be unreasonable to expect a definitive theoretical model to be presented here. That's not how discovery works. However, in cases like this, especially for a high impact journal like Nature Communications, the authors should be expected to canvas the literature carefully to consider all feasible hypothesis. In this paper, the authors are not considering some known physical effects, and some recently proposed ones, that could be strongly influencing their results. This is my first major concern. In addition, the model they do present, is not clearly presented. I was not able to understand what the main physical effect(s) they claim are at play. This is my second major concern. I elaborate on these points below.

Regarding the first major concern (other possible effects), the authors do not cite recent work from Laurie Marks' group linking flexoelectricity with triboelectric charge transfer, e.g.:

10.1063/5.0048920

10.1103/PhysRevMaterials.5.064406

<https://doi.org/10.1021/acs.nanolett.2c00240>

10.1103/PhysRevLett.123.116103

<https://doi.org/10.1021/acs.nanolett.2c00107>

I have read much of that work and find it sufficiently convincing (although at times overstated in its generality) that it merits consideration. While the authors here are not considering charge transfer itself, if the rate of charge transfer from the sample surface to the tip depends on the polarization, then the tip will have on average more charge on it when sliding over up domains vs. down domains. More charge will then lead to more electrostatic attraction between the tip and sample in addition to the applied normal load (effectively, increasing the tip-sample adhesion), increasing the total load, and thus increasing the contact pressure. Increased contact pressure is almost always correlated with higher friction and wear. The authors may argue that charge accumulation on the tip should not be expected with the conductive tip, but if there are any defects that can trap charge, or an oxidized outer layer on the tip, that could still lead to this effect.

Another possible effect is that there will always be a contact potential difference between the tip and sample (whether the tip is conductive or not). Taking the tip's work function as fixed, the contact potential will be different between up and down domains. This again will lead to different total normal loads, higher pressures, higher friction, and more wear.

The authors exclude an electrostatic switching effect by running tests at voltages ranging from -150 to +150 volts, but this is an abnormally large voltage to apply to an AFM tip. This often can cause dielectric breakdown, large polarization forces, and other difficult-to-control effects. The authors should show what happens at more modest voltages, like -10 to +10 V, where they are likely to cross over the contact potential difference between the tip and sample. They can also measure or at least estimate the contact potential difference with Kelvin Probe force microscopy, or simply by measuring the deflection of the tip as the voltage is ramped from -10 to +10 V while the tip hovers close to, but not in contact with, the surface.

They could also have checked for differential adhesion by measuring the pull-off (adhesion) force between the tip and sample. This is a very routine measurement. Maps of adhesion over the different domains is thus necessary for this paper. The authors would also benefit from measuring the friction force as a function of load, down to the pull-off force, which can give a better sense of the contact mechanics at play.

On other possible aspect to consider is the role of mechanochemistry in wear. Stress-induced acceleration of wear through an Eyring-like model, based on Arrhenius kinetics, has been demonstrated in several cases with AFM. The idea is that atomic-scale wear involves breaking a bond (or bonds) between the atom being removed and the other atoms in the solid to which it was coordinated. This is a chemical process, and Eyring and Bell, showed that this can be described by a reduction of the energy barrier for the bond breaking processes that depends linearly on the applied force (Bell model) or stress (Eyring model). Could an electric field be coupled in to this framework? Perhaps something could be determined by analogy to electric field effects in chemical reactions.

Regarding my second major concern (unclear explanation of their model): The authors do not explain the essential physical mechanism underlying their explanation for the observed effect. They say in the manuscript “flexoelectrically induced polarization and the consequent asymmetry of mechanical properties between oppositely polarized up and down domains.” What mechanical properties are they referring to? Elastic moduli? Yield strengths? Fracture toughnesses? Moreover, the statement above is not clear or specific; the details are left to a model described mostly in Methods and Supplementary Text, which is difficult to follow and filled with mechanics jargon. The authors need to clearly lay out what the mechanism is.

They also share – to their credit – the unusual observation that the friction contrast changes depending on the slow-scan direction of the AFM image. This is very unusual, and their explanation is that “the height and friction differences signals (Fig. 1h) oscillate because of the contact geometry difference between frame-up (scan from below to above) and frame-down (scan from above to below) during the continuous, repeated milling scans.” This doesn’t cut it. What evidence to they have that the contact

geometry changes with the slow scan direction? Why would it? Does the magnitude and sign of this difference change if different tips are used? It is likely that something else is going on, and it may be clue regarding the mechanism at play, perhaps related to the displacement of adsorbates on the surface by the scanning action tip, or perhaps cross-talk between normal and lateral signals of the cantilever/photodiode. I don't have an explanation, but it's odd, and the effect is strong. Note that only the friction signal oscillates; the height differences does not, although the etch rate (derivative of height with scan number) does appear to oscillate.

Finally, going back to the comment above about the "asymmetry of mechanical properties": the authors themselves (to their credit) note that "this asymmetry implies an anomalous, positive correlation between the hardness of the domains and their etch rate, while normally high hardness materials are more resistant to wear as reported in the form of Ashby plot between hardness and wear coefficient." It's honest of them to point this out, but doesn't it suggest that the model may not be correct?

In short, the authors need to shore up their and better explain model, consider other possible mechanisms (including ideas presented in literature that were not cited), and explain the unusual slow-scan direction observation.

Finally, a few minor observations:

Abstract: I don't like "Artificially induced" since it's not artificial; perhaps they mean "Externally induced" or just "Switchable"

Abstract: the fact that a diamond tip was used should be mentioned; wear depends on both materials in contact, not just one.

Abstract: the last sentence seems a bit exaggerated, especially the use of "establish".

Lines 46-47 "friction and wear coefficients" are empirical quantities. "Shear strength" can just be used instead of "friction coefficient" for example. Wear is trickier but I don't believe the "wear coefficient" is something they are controlling.

Lines 50-51: "...no previous studies..." This is a rather sweeping statement. Are the authors sure? The final sentence of the same paragraph is more measured.

Line 64: I believe the authors mean to say "... is asymmetric and *the existence of the asymmetry* is independent of surface chemistry" since the magnitude of the asymmetry surely varies with the surface chemistry.

Line 91: "polarization-dependent tribology". Since tribology means friction, wear, adhesion, and more, what specifically are the authors referring to here? I think they mean to say "polarization-dependent friction and wear"

Line 170: "Because friction strongly depends on the real contact area, as does the mechanical wear rate," seems off. It would make more sense to say that, at higher applied normal stresses (due to higher normal load), one gets more contact area (and thus friction force), and one gets more wear because of higher normal stresses (and perhaps also because of the higher friction forces).

Response to Reviewers

For convenience of the editor and referees, comments provided by the editor and reviewers are presented in blue and our responses are inserted in black. In the manuscript, the modifications are highlighted in yellow.

Reviewer 1

Overall comments: This paper has a large amount of excellent science. Unfortunately it is also a very hard read, and will confuse most people, reducing its impact. The most obvious problem is that it has 23 pages for the main text, and 53 for the Supplemental, with most of the hard science in the SM.

In my opinion this work should be split into two papers which are related, but are fundamentally different:

Paper 1: Analysis of the coupling between FeX and ferroelectricity. This would include the theory and parts of the experiments. The authors would need to demonstrate reasonable agreement between the theory and experiments, which is not completely clear.

Paper 2: The work involving wear to produce different shapes etc.

By splitting the paper it will become much, much more readable. In addition, the authors can provide much more useful information to the larger community. While the two papers could, perhaps appear in the same journal, there are reasons to consider the wear paper appearing in a more tribology related journal. Even though the science is very high quality, I cannot recommend the paper for further processing in its current form.

Overall Response: We sincerely appreciate the reviewer's time and effort spent evaluating our manuscript and providing positive comments. We agree with the reviewer that our initial manuscript contains a very daunting amount of data. We note, however, that the application discussion follows naturally as the final aspect of a unified storyline: description of first

observation, studies of general characteristics, investigation of mechanism, and applications of newly observed phenomena. Therefore, we would like to present the data in a single publication. We attempted to streamline the manuscript. In addition, we believe that patterning application results presented in our manuscript will be of interest to a large community of researchers, potentially well beyond simply the community interested in fundamental aspects of ferroelectrics. Therefore, we retained figures from patterning results, but some texts have been shortened in the discussion.

Comment #1: Are there differences in fracture toughness depending upon the polling direction? This matters for wear.

Response #1: We appreciate the comment from the reviewer. Flexoelectricity significantly influences the fracture toughness of ferroelectrics. Previous studies have theoretically¹ and experimentally² shown that the fracture toughness is asymmetric depending on the polarization direction. The theoretical study¹ demonstrated that the fracture toughness is asymmetric with respect to the sign of polarization. Furthermore, experimentally, crack propagation is shown to be asymmetric depending on the polarization orientation².

However, we argue that the fracture toughness is not dominant during asymmetric wear in our presented data (e.g., Fig. 1d). If the fracture toughness matters in our milling results, the surface roughness should increase after milling scans, however we rather see the decrease of surface roughness after the milling scan (We also thanks the reviewer's important Comment #16). Compared with pristine surface, the surface roughness for up domains decreases from 207.228 pm to 198.667 pm, and 224.681 pm to 101.669 pm for down domains (Please see Fig. R1). We also have added Fig. R1 in Supplementary Information (Fig. S4).

Fig. R1. Surface roughness before (a) and after (b) milling scans in Fig. 1. Orange color represents the region acquired for the roughness in up domains, while purple indicates the region for the roughness in down domains. The roughness decreases after milling scans in both polarization domains.

Comment #2: Please don't use the term "etching" anywhere, that has an accepted usage with solvents.

Response #2: The referee raised an important point. We have modified our manuscript and supplementary information in accordance with the reviewer's suggestion, which we greatly appreciate. The term "etch" has been replaced by "wear", "grinding", "polishing" or "patterning". In the revised manuscript, the modifications are highlighted in yellow.

Comment #3: An important experiment would be to use tips of different radii to prove that FeX matters. The authors could also analyze results for different silica, rather than just saying that comparable sizes work. (The force applied will also matter.)

Response #3: We conducted additional experiments in response to the reviewer's insightful comment. We note that the milling experiment is highly sensitive because of the complex interaction between probe, wear debris and surface contamination. Therefore, it is sometimes challenging to quantitatively compare the results. Despite the challenges, we conducted additional experiments using probes with similar cantilever geometry and spring constant, but with different nominal radius of tip (Table R1). As can be seen in Table R1, the nominal radius of pristine CDT-

NCHR (*Nanosensors*) is 7.5 times larger than that of NC-LC (*Adama Innovations*). Using the two probes in separate experiments, we conducted 4 continuous milling scans in different regions from the loading force of 1 μN to 10 μN respectively (Fig. R2). Fig. R2 depicts resulting topography images after milling scans, line profiles from the height images, and reference PFM phase images. More specifically, Fig. R2a and Fig. R2b depict surface topography images after milling scans in each region with different loading forces. In Fig. R2c and Fig. R2d, for more detailed look at milling results, we investigated the line profiles of height from 1 μN to 5 μN regions in each case. We observe no significant wear of the surface with either probe under a load of 1 μN . However, under a load of 2.5 μN , asymmetric wear starts in NC-LC case (smaller tip radius) with discernible height difference between up and down domains, which further increases under a load of 5 μN . In CDT-NCHR case (larger tip radius), we observe some wear of the surface, but with no asymmetry at 2.5 μN , and only in the regions scanned with a load of 5 μN do we start to observe the initiation of asymmetric wear. These results suggest that flexoelectricity can matter because the wear asymmetry in the case of larger strain gradient (smaller radius tip) starts at lower force than in the opposite case. This is one of the features of flexoelectricity; the flexoelectrically induced response is greater at smaller scale. In addition, the wear depth (or wear volume) is greater for a probe with smaller radius, which implies that the stress concentration on the surface plays an important role in asymmetric wear of ferroelectrics.

Probe	Length (μm)	Thickness (μm)	Width (μm)	Spring	Tip radius
				constant (Measured, N/m)	(nm)
NC-LC (Single crystalline diamond)	125 ± 10	4 ± 0.5	40 ± 5	54.848	20 ± 10
CDT-NCHR (Conductive diamond coating)	125 ± 10	4 ± 1	30 ± 7.5	64.375	150 ± 50

Table R1. Comparison of NC-LC and CDT-NCHR specifications.

Fig. R2. Milling experiments depending on the radius of probe. (a) Height after four continuous milling scans at loading force from 1 to 10 μN using a NC-LC probe. (b) Height after four continuous milling scans at loading force from 1 to 10 μN using a CDT-NCHR probe. Four continuous milling scans were conducted in each region. (c) Height line profiles from (a) (1 μN , 2.5 μN and 5 μN regions). (d) Height line profiles from (b) (1 μN , 2.5 μN and 5 μN regions). Line

profiles were acquired along the domain boundary. (e) PFM phase after milling scans in the NC-LC case. (f) PFM phase after milling scans in the CDT-NCHR case.

Comment #4: How do their friction and wear anisotropies compare to other work in the literature?

Response #4: We greatly appreciate the reviewer's comment. Previous work³ (already cited in the manuscript) on friction imprint of ferroelectric single crystalline BaTiO₃ reported a friction asymmetry depending on polarization orientation. However, in the previous study³, the applied loading force is only 12 nN, which is significantly less than in our case. In this force regime, they observe no surface wear during the scan. In addition, surface screening conditions and adsorbate effects dominate friction asymmetry, as they observed different friction behavior after surface modification by heating and cooling the sample. However, in our case (μ N scale loading force for asymmetric wear), flexoelectrically induced friction asymmetry dominates the friction signals and the observed trend for the asymmetry is insensitive to temperature change (Fig. R3 in response #5), but highly sensitive to applied loading force and probe degradation. Therefore, we claim that our work has its own strong novelty and uniqueness.

Comment #5: What is the mechanism of what they call "etching"? Just calling it wear is not enough. Is it similar to atomistic on layer-by-layer wear, see for instance DOI 10.1080/09506608.2016.1213942. Is there ploughing?

Response #5: The reviewer raised very important point. We argue that the mechanism of asymmetric wear in our manuscript is very similar to that of layer-by-layer wear^{4,5}, but not comparable to the mechanochemistry⁶ for the observed tribological asymmetry. We carried out further experiments and analyzed our data, as detailed below to describe the mechanism.

First, in our case, thermal activation is not dominant in the observed tribological asymmetry of ferroelectrics. The atom-by-atom wear is a stress-assisted chemical reaction, and in this mechanism, temperature as well as mechanical stress is an important factor because it follows Arrhenius-type reaction kinetics⁶. If the wear asymmetry mechanism follows the mechanochemistry mechanism proposes in this reference, we expected the wear volume to increase with increasing temperature at a fixed load. We designed the milling experiments at optimized loading forces with increasing and decreasing temperature shown in Fig. R3 (updated

in the Fig. S25), as we had previously observed the effect of degradation (Figs. S11 and S12). Fig. R3 shows height and PFM phase after milling scans depending on the environment temperature. We used a single diamond probe, and conducted milling scans at 0°C and then performed milling scans at increased temperature of 90°C (Fig. R3a). As the temperature increases from low to high, we observe the decrease in wear depth. Ramping the temperature back down from 90°C to 0°C, the wear depth also shows decreasing behavior (Fig. R3b). We conclude that temperature is not the dominant factor in our case; however, loading force and degradation of the probe (especially attachment of wear debris) are the critical factors for asymmetric wear. Therefore, we exclude atom-by atom wear or mechanochemistry⁶ as a mechanism for tribological asymmetry.

Second, the mechanism behind the asymmetric tribology is strongly dependent on the applied stress to the sample surface. From the observed features of tribological asymmetry (Figs. S7–12), the important factors for asymmetric wear are loading force and tip condition, which can influence the normal/shear stress at the junction of tip and sample during sliding or ploughing. Furthermore, even under same milling loading forces, after a contact scan performed well below the milling force could change the friction and wear behavior (Fig. 1h). This is also because the wear debris distribution between probe and sample can be altered to change the interface mechanics during the single contact scan. Additionally, the surface roughness decreases after the milling scans. These results support the layer-by-layer wear mechanism for the tribological asymmetry.

Fig. R3. Temperature-dependent milling experiment on PPLN. (a) Height after milling scans from 0°C to 90°C including the worn regions at the loading force of 10 μN. (b) Height after milling scans including the worn regions from 90°C to 0°C at the loading force of 11.5 μN using the same probe used in (a). In the milling experiment, a single probe (NM-TC) was used and the loading force was slightly increased from 10 to 11.5 μN for comparison at the optimized condition. (c) PFM phase after milling scans including the worn regions from 0°C to 90°C. (d) PFM phase after milling scans including the worn regions from 90°C to 0°C.

Regarding ploughing of sample by AFM tip, we note that we do observe multiple cases with a broad spectrum depending on types of the probe, loading forces and samples. For one extreme, the wear depth is approximately 5 nm after a single scan (Fig. 2h–j), which is a multiple layer wear per scan. In this case, we expect that ploughing can occur because the wear is quite severe.

For the other extreme, we also observe the tribological asymmetry with a Pt/Ti coated Si probe in PPLN. Young’s modulus of Pt, Ti and Si are expected to be lower than that of LiNbO₃, so ploughing should not be dominant. However, we also observe the tribological asymmetry in this case (Fig. R4).

With these two observed extremes, it will be fascinating to conduct detailed research that includes in-situ observation of wear to observe the dominant effect (either wear by high junction stress or

wear by ploughing). However, we believe that this detailed study is beyond our current work, but we plan to conduct the future study for in-situ observation of wear in collaboration with other research groups (e.g., Prof. Laurence Marks group at Northwestern University).

Fig. R4. Tribological asymmetry with a metal-coated probe. (a) Topography and (b) PFM phase after the multiple milling scans using a Spark 350-Pt probe. (c) Friction image during the milling scan showing higher friction in up domains and lower friction in down domains.

Comment #6: On p6, line 158 the statement about adsorbates is only right if the pressure is low enough – remember 1 monolayer in 10^{*-6} seconds. Similarly, “scraping off screening charges” seems inappropriate, as electrons move really fast.

Response #6: We appreciate the keen comment. In the statement about adsorbates, we initially considered the possible asymmetric adsorption on pristine sample depending on polarization configuration. Indeed, we found the literature that showed exemplary selective adsorption on ferroelectric surface⁷. Regarding the statement of screening charges, we agree with the reviewer that it can confuse readers, so we’ve modified our statement like below.

Before: However, if the friction properties are indeed governed by screening charges (electrons or holes) or adsorbates (especially chemisorbed species), the asymmetry should disappear with continuous milling scans, as any surface species or asymmetric skin layers are gradually removed. This is not the case in our results. Rather we find that the asymmetry persists throughout the full cycle of continuous etching.

After: However, if the friction properties are indeed **governed by screening conditions or adsorbates (especially chemisorbed species)**, the asymmetry should disappear with continuous milling scans, as any surface species or asymmetric skin layers are gradually removed. This is not the case in our results. Rather we find that the asymmetry persists throughout the full cycle of continuous **milling**.

Comment #7: In S5 how are the values for the strain-gradient terms calculated? There is one estimate by Stengel, but I do not see any values in the paper.

Response 7: We appreciate the reviewer's comment. There is no reliable experimental estimate of the strain-gradient material parameters. Ab initio estimates are not necessarily representative of actual materials. In our and other groups' experience, within reasonable values, this parameter has limited impact on the results. However, if too small, the mathematical problem becomes unstable. In our calculations, we took this parameter to be a bit larger than required for mathematical and computational stability. In the revised version of the manuscript, we have simplified the section, but added details including strain-gradient term calculation.

Comment #8: In S5 what boundary conditions are being applied on the free surface of the ferroelectric? As written the equations may be clear to the authors, but not to the general audience. Most readers will have no idea (I don't) why a saddle-point problem is needed, and whether the "weak form" means anything.

Response 8: We thank the reviewer's comment and apologize for the inadequate tone of S5 in our original manuscript. In the revised manuscript, we have rewritten this section and removed from our manuscript the description of saddle-point problem and weak form. We have eliminated superfluous equations and statements, and clearly described the boundary conditions considered

in our calculations. As clearly stated in the revised supplement, we impose open circuit conditions by prescribing the electric potential at the bottom and leaving it free at the top. Furthermore, regarding the boundary conditions of ferroelectric, we impose open circuit conditions by prescribing the electric potential at the bottom and leaving it free at the top. We conducted additional simulations and we have examined the sensitivity of our main finding (the asymmetry of the response depending on polarization direction) to the electrical boundary conditions by considering short-circuit conditions. These results are shown in Fig. R5 and Fig. S22.

Fig. R5. Electric potential distribution and indentation depth depending on electric ground condition. (a–c) Electric potential distributions upon indentation using the spherical AFM probe with different grounding conditions. Electric potential distribution considering the bottom of sample is ground (a). Electric potential distribution considering the bottom of sample and contact zone are ground (b). Electric potential distribution considering the bottom of sample, surface of sample and contact zone are ground (c). (d–f) Simulated indentation depth for (a–c), respectively. Orange color is used for up domains and purple color is used for down domains.

Comment #9: How large are the coupling terms? It would be of general importance to know whether the FeX and strains can be considered as additive, or there are large coupling terms so a fully self-consistent solution is needed. From Fig. S20 they do not look large.

Response 9: The asymmetric response depending on polarization orientation originates from the competition between the mechanical effects of piezoelectricity and flexoelectricity. The asymmetry depends on the strength of the flexocoupling terms as shown in Fig. S21 (Fig. S20 in previous manuscript), if piezoelectric parameters are fixed. If the problem were approached in a non-self-consistent manner, then there would be no difference since the input and the output are both mechanical (the electric problems would be one-way coupled). In our calculations, the solution of the electro-mechanical problem is self-consistent. We did not examine the accuracy of simplified models, which should be problem-dependent.

Comment #10: In S6 why is the nominal value from the manufacturer of the tip used? Under conditions where there is wear of the substrate one expects wear of the tip.

Response 10: Thank you very much for your comment. In the perspective of simulations, our main goal is not to focus on quantitative analysis, but rather to provide evidence supporting a specific mechanism. Here, we estimated that flexoelectricity can matter both in simulation and experiment where the indentation or contact mechanics could be different depending on the polarization direction. The realistic value of the probe radius during the milling scan varies significantly.

Comment #11: The model in S6 the authors use appears to reduce to a standard Hertz model, where the FeX contributions have already been analyzed by Mizzi et al and others more recently in the mechanics literature. Indeed, the plot they show looks just like a classical Hertzian solution. It is important to know if there results are the same or different. Currently this other work is not cited.

Response #11: We value the reviewer's comment and indeed we used a standard Hertz model, which is similar to previous studies⁸⁻¹⁰. We've updated the reference in the manuscript. However, we think that our work and the previous studies are similar in the sense that a probe indented on a surface and the flexoelectric effect is invoked, but other than that, the rest is quite different. They

investigated triboelectricity by examining centrosymmetric non-piezoelectric materials, whereas we focused on the interaction between piezo- and flexoelectricity as the key to our proposed mechanism to explain the observed asymmetry.

Comment #12: In S8 the statement that the Pt/Ir probe did not yield asymmetric wear (don't use etching please) needs explanation. This is a very interesting result.

Response #12: We thank the reviewer for the comment. We are sorry for the confusing expression in the initial manuscript. We didn't observe the significant surface wear, thus no asymmetric wear using the Pt/Ir coated probe (EFM, *NanoWorld*, $k \sim 2.8$ N/m), because the maximum loading force with the probe is not enough to initiate asymmetric wear.

We conducted the additional experiment using a metal-coated probe with higher spring constant (Pt/Ti coated Si probe, Spark 350-Pt, *NuNano*, $k \sim 42$ N/m). In this case, we do observe the asymmetric tribology as shown in Fig. R4. The summarized table for probe selection for the asymmetric wear of PPLN is shown in Table R2. Therefore, if the applied stress is enough to initiate the wear asymmetry, metal-coated probes could also be used, even if the wear rate is much lower than the diamond probes.

Probe	Description	Spring constant (N/m, nominal)	Asymmetric wear of PPLN
NM-TC	Single crystal diamond (conductive)	350	Observable
DT-NCHR	Diamond-coated (conductive)	80	Observable
CDT-NCHR	Diamond-coated (conductive)	80	Observable
EFM	Pt/Ir-coated (conductive)	2.8	Unobservable (No surface wear)
HQ:NSC16/HARD/Al BS (DLC-coated)	Diamond-like-carbon-coated (weakly conductive)	40	Observable
D300	Single crystal diamond (non-conductive)	40	Observable
NC-LC	Single crystal diamond (conductive)	125	Observable
Spark 350-Pt	Pt/Ti-coated (conductive)	42	Observable

Table R2. Probe selection for the asymmetric wear of PPLN.

Comment #13: In S9 the statement at the end needs to be “the presence of pre-existing defects”. Wear is due to defects, often dislocations or fracture.

Response #13: We thank the reviewer for the comment. We have modified the phrase like below in response to the reviewer’s comment.

Before: Therefore, we conclude that the presence of defects is not responsible for the observed asymmetry.

After: Therefore, we conclude that the presence of **pre-existing** defects is not responsible for the observed asymmetry.

Comment #14: In S10 the strain fields for different tips and hence the coupling is very different – this needs to be mentioned.

Response #14: We thank the reviewer for the detailed comment. We have added the sentence in the revised manuscript like below.

The strain fields with two different indenter (spherical and Berkovich) are expected to different because of the tip geometry is different; however, we assume that Berkovich indenter generates a stronger strain field than spherical indenter.

Comment #15: In S12 and elsewhere, what is the wear rate? Are we dealing with multiple layers per pass or what? Based upon Fig S2 it looks like at least a monolayer per scan.

Response #15: The reviewer raised a very significant point. The wear rate is strongly dependent on the types of probe, loading force, probe condition and sample. According to the data presented in Fig. S3 (tip: NM-TC (single crystal diamond, sample: PPLN, loading force: 5 μ N), wear rates are 0.0466 nm/pass for up domains, 0.0364 nm/pass for down domains. The values were obtained simply dividing the wear depth by 100, which accounts for total number of scans multiplied by two that include both trace and retrace motions. Furthermore, from the PbTiO₃ case (tip: 4XC-GG (metal-coated), sample: PbTiO₃ thin film, loading force: 1.6 μ N, the total number of milling scans is greater than one hundred, and the milling depth is around three unit-cell. We note that wear rates in these cases are underestimated due to the degradation and the mechanical softness of the tip, but these values are less than a monolayer wear per scan.

For the other extreme case, the case from Fig. 2i (tip: D300 (single crystal diamond), sample: PPLN, loading force: 6 μ N), the wear depths are 5.84 nm for up domains and 5.32 nm for down domains after a single milling scan. Therefore, the wear rates are 2.92 nm/pass for up domains and 2.66 nm/pass for down domains. In this case, multiple layers of ferroelectric oxide are worn.

Therefore, we claim that the wear rate can be strongly influenced by probes (tip condition, cantilever geometry, materials of tip), applied stress (loading forces, tip radius) and samples (thickness, mechanical properties, dielectric permittivity of sample).

Comment #16: What is the roughness of the surface after the wear? This matters to know if we are talking about fracture or something different.

Response #16: We appreciate the comment and we think that response #1 adequately addresses this comment.

Comment #17: What is “a.u.” in Figure S3d?

Response #17: We thank the comment from the reviewer. “a.u.” in Fig. S3d and other friction images at high forces is an arbitrary unit for the friction data, because the calibration for friction forces is challenging due to continuous degradation of tip surface during the milling scans. It will be interesting to conduct more quantitative analysis of asymmetric friction forces during continuous milling scans. We plan to conduct the study in the future but we think that it is beyond the scope of the current work.

Comment #18: Why does Fig S18. Have polarization in the vacuum?

Response #18: We appreciate the reviewer’s insightful comment. The polarization field depicted in Fig. S18 (now Fig. S19) represents the induced polarization at the starting point of polarization vector. We agree with the reviewer that it may confuse the reader. However, to compare the piezoelectric polarization with the flexoelectric polarization with smaller amplitude, we initially illustrated the figure as it appeared in the original manuscript. To prevent the confusion, we have added the following explanation.

The arrows represent the induced polarization at the starting point of polarization vector.

Comment #19: What is the surface charge on the diamond tip?

Response #19: We value the comment from the reviewer. We are not very sure about the precise surface charge species on the diamond probe, and surface charge species and concentration on the diamond tip, which originated from the local defects in diamond probe, may change the trend of tribological asymmetry. However, we claim that we do observe the tribological asymmetry regardless of surface charge state of the tip.

First, experimentally, the observed tribological asymmetry arises with or without the external DC bias, which can alter the surface charge states. During the milling scans, both tip and sample are electrically ground in most experiments. We further proved that there is no significant change in asymmetric tribology even with high voltage application to the tip (Fig. 2k–n). In response to Reviewer 3’s comment #3, we also conducted milling experiment with moderate voltage application (from -10 V to +10 V), but the observed trend is the same; Friction and wear rate is higher in ferroelectric up domains than down domains, and external bias does not significantly change the trend.

Moreover, we further conducted the milling experiments with various types of tip. Those tips consists of various materials (metal-coated Si probe or diamond probe), and the tips exhibits different electric conductivity (Fig. S23 in the revised Supplementary Information). Fig. S23 depicts the probes have different electric conductivity (highly conductive, weakly conductive or non-conductive). Regardless of the substance of tip and conductivity of probe, under high stress application, we observe the same asymmetrical trend (Table R2). For example, metal-coated tip (Spark Pt-350) show same asymmetry trend as already shown in Fig. R4 as compared to and diamond tips (NM-TC, NC-LC, CDT-NCHR, DT-NCHR, D300). Additionally, conductive single crystalline diamond tip and insulating diamond tip do show the same asymmetric trend, where the ferroelectric up domains exhibit higher friction coefficient and wear rate.

In addition, we also conducted additional simulations with different ground conditions as shown in Fig. R5. However, we observe that the different ground condition shows various electric potential distribution (Fig. R5a–c), the indentation depth is always higher in ferroelectric up domains (Fig. R5d–f). Therefore, we claim that the surface charge state is not dominant in the trend of asymmetric tribology, as we have proved both experimentally and theoretically.

Comment #20: I am dubious about the relevance of Fig S21.

Response #20: We appreciate the comment and believe that our response #19 adequately addresses this comment.

Comment #21: The “TiO₂” terminated SrTiO₃ substrate appears in Fig S28 without any other mention. There is also nothing about how this was prepared – I assume it is some double-layer from buffer etch+anneal, but that needs to be stated. I am not sure about the relevance of Fig. S28.

Response #21: We thank the comment from the reviewer. The TiO₂ terminated SrTiO₃ used in the experiment is the sample that we purchased from *CrysTec GmbH*. We did not performed special cleaning, etching and annealing process, but conducted growth and characterization. As depicted in Fig. R6, the TiO₂ terminated SrTiO₃ substrate shows beautiful unit-cell terrace structures, as opposed to the half unit-cell terrace structures¹¹ in the mixture of SrO and TiO₂ terminated SrTiO₃ substrate. After the growth, in the pristine PbTiO₃ surface, terrace structures could not be observed (Fig. S31), however, after appropriate milling scans, we observe the unit-cell terrace structures as can be seen in Fig. R6b. These results suggest that the unit-cell resolution lithography (out-of-plane scale) is possible. We have updated with additional explanation and modified the figure in the revised version.

Fig. R6. Lattice-scale polarization-derived lithography. (a) Surface topography of TiO₂ terminated (001)-oriented SrTiO₃ substrate and (b) **patterned** PbTiO₃ thin film showing atomistic terrace edge features in the bottom of figure. (c and d) Line profiles along the terrace edge in (a) and (b). Scale bars are 300 nm.

Comment #22: In Table S1 they state isotropic FeX – that is very dubious, although it may not be far off. Remember that the FeX coefficient for STO are different for [001] and [110].

Response 22: We thank to the reviewer for the keen comment. Given the uncertainties in our flexoelectric parameters, we chose an isotropic model with only two independent ones. However,

we think that the isotropic model can also show the effect of flexoelectricity in the observed tribological asymmetry.

Reviewer #2

Comment: This manuscript makes an excellent point, artificially induced asymmetric tribological properties of ferroelectrics, that offers an alternative route to visualize and control ferroelectric domains. The switchable friction and wear behavior of ferroelectrics using a nanoscale scanning probe can be used as smart masks, and the asymmetry is enabled by flexoelectricity coupled polarization in the up and down domains under a sufficiently high contact force. The polarization-sensitive tribological asymmetry is widely applicable across various ferroelectrics with different chemical composition and crystalline symmetry. And using this switchable tribology and multi-pass patterning with a domain-based dynamic smart mask, this manuscript demonstrated three-dimensional nanostructuring exploiting the asymmetry wear rates of up and down domains. The findings establish that ferroelectrics are electrically tunable tribological materials at the nanoscale for versatile applications. I hence recommend it for publication.

However, the authors should probably consider the following issue before producing a final manuscript:

The asymmetric tribological properties in this manuscript were demonstrated in PPLN (periodically poled lithium niobate), in which the polarization in the adjacent domains were up and downward aligned. Would the authors verify the asymmetric tribological properties in other ferroelectrics, where the polarization might be rotated and be not distributed up and downward in the adjacent domains?

Response: We appreciate the reviewer's positive comments. Regarding your insightful comment on the in-plane polarization effect on the tribological asymmetry, we would like to explain your comment using results of the large scale patterning of PZN-PT (001) single crystal.

The scalable polarization-derived lithography of PZN-PT single crystal is depicted in Fig. R7. Point 1 and point 2 in Fig. R7 represent domains with the same vertical PFM phase (Fig. R7c and R7f), but distinct vertical PFM amplitude (Fig. R7b and R7e). This result indicates that the two domains do have two different in-plane polarization. However, the topography image after polishing with silica particles (Fig. R7a and R7d) show no observable height difference between point 1 and point 2. Therefore, we conclude the in-plane polarization effect is not dominant in the ferroelectric PZN-PT single crystals. We thank again for the reviewer's positive comment.

Fig. R7. Scalable polarization-derived lithography of PZN-PT single crystal. (a) Height, (b and c) vertical PFM amplitude and phase after polishing using silica nanoparticles for 20 min. (d) Magnified Height, (e) vertical PFM amplitude and (f) phase. Number 1 and 2 indicate down domains but with different in-plane polarization direction. Scale bars are 1 μm for (a–c) 400 nm for (d–f).

Reviewer #3

Overall comment: The manuscript by Cho et al presents a novel and impressive new method to pattern materials at small scales by exploiting a mechanism that itself is newly discovered: the dependence of the wear rate of a ferroelectric material on its local polarization orientation. In particular, regions of ferroelectrics poled “up” or “down” (with respect to the surface normal) are both worn when a diamond AFM tip is slid in contact over it under ambient conditions, but the wear occurs at different rates. This produces a height difference between regions of up vs. down polarization. The authors show the effect occurs between a minimum and maximum range of applied loads, and leads to height differences of a few nm before leveling off. A difference in friction is also observed between up- and down-polarized domains. The authors then show numerous ways the effect can be exploited for characterizing the orientation (up vs down) of previously poled ferroelectric domains; maskless patterning; and tomographic probing. They also show how polishing with a slurry of small particles can also produce the same effect, but across an entire wafer, in their words, “simply polishing the whole crystal using silica nanoparticles that effectively act as millions of mobile AFM tips.”

The results are innovative and interesting, and highly novel. The authors present multiple examples using different piezoelectric materials, demonstrating the effect is reproducible and can be well-controlled. The experimental skill required to conduct this study is substantial.

Overall response: We appreciate the reviewer’s time and effort spent in evaluating our manuscript, as well as positive comments on the concept of our work. We strongly believe that our work presents innovative research results of broad significance that offer a fascinating playground for studies of ferroelectrics and related correlated oxide systems. We are confident that many research groups will soon apply the presented findings and develop them further.

Comment #1: The authors present a finite element model to help explain the results. However, the main shortcoming of the paper is that the physical mechanism behind the effect is not well-explained, and other possible mechanisms (discussed below) are not considered. To be sure, this is a very complex system, as it involves: a nanoscale contact, where the stresses will be non-uniform and may even deviate from continuum predictions; the combined and interacting effects of flexoelectricity and piezoelectricity; and the very complex challenge of understanding friction and wear and their interaction, for which few robust, physical-based theories exist even for simple materials, let alone the complex ones studied here which involve multiple coupled physical phenomena.

So, it would be unreasonable to expect a definitive theoretical model to be presented here. That's not how discovery works. However, in cases like this, especially for a high impact journal like Nature Communications, the authors should be expected to canvas the literature carefully to consider all feasible hypothesis. In this paper, the authors are not considering some known physical effects, and some recently proposed ones, that could be strongly influencing their results. This is my first major concern. In addition, the model they do present, is not clearly presented. I was not able to understand what the main physical effect(s) they claim are at play. This is my second major concern. I elaborate on these points below.

Regarding the first major concern (other possible effects), the authors do not cite recent work from Laurie Marks' group linking flexoelectricity with triboelectric charge transfer, e.g.:

10.1063/5.0048920

10.1103/PhysRevMaterials.5.064406

<https://doi.org/10.1021/acs.nanolett.2c00240>

10.1103/PhysRevLett.123.116103

<https://doi.org/10.1021/acs.nanolett.2c00107>

I have read much of that work and find it sufficiently convincing (although at times overstated in its generality) that it merits consideration. While the authors here are not considering charge transfer itself, if the rate of charge transfer from the sample surface to the tip depends on the polarization, then the tip will have on average more charge on it when sliding over up domains vs.

down domains. More charge will then lead to more electrostatic attraction between the tip and sample in addition to the applied normal load (effectively, increasing the tip-sample adhesion), increasing the total load, and thus increasing the contact pressure. Increased contact pressure is almost always correlated with higher friction and wear. The authors may argue that charge accumulation on the tip should not be expected with the conductive tip, but if there are any defects that can trap charge, or an oxidized outer layer on the tip, that could still lead to this effect.

Response 1: We appreciate the reviewer's constructive comment. We regret that our initial literature review was somewhat insufficient and thank the reviewer for suggesting alternative mechanism to examine. First, we reviewed recent works from Prof. Laurence Marks' group, which reported triboelectric charge transfer using flexoelectricity⁸⁻¹⁰ and revised our manuscript with updated citations and additional text. We agree with the referee that charge transfer during sliding of the probe over the oppositely polarized surface may induce asymmetry due to the potential difference during scanning. From the theoretical study⁸, flexoelectrically induced potential difference during indentation and pull-off generates approximately 6.5 V. The scanning process consists of repetitive indentation and pull-off of AFM tip in one/two-dimensional motion; therefore, it is necessary that the charge transfer mechanism be examined.

If we assume the charge transfer mechanism is true, flexoelectrically induced charge will also matters as evidenced by colossal flexoresistance¹² or flexoelectronics¹³ of centrosymmetric materials. This is what happens when a flexoelectrically induced current is generated via indentation; either holes are transported to the tip or electrons are transferred away from the tip. Then, on alternately poled ferroelectric surface, if we further assume that the triboelectric charge transfer to the probe occurs, positive screening charges will be easily collected on the down domains or negative screening charges will be easily gathered on the up domains. In combination with flexo-induced current, less charges are collected on up domains and more charges are collected on down domains. This is opposite to our continued observation and the reviewer's hypothesis because we observed the higher friction and severe wear on up domains than down domains.

In addition, as already described in Fig. 2 of the manuscript, we were unable to observe any substantial difference between up and down domains under voltage application ranging from -150 V to 150 V while applying high strain gradient to the sample. This result indicates that the charge transfer mechanism is not dominant in our case. The applied voltage range is not high enough for

single crystal ferroelectrics and is still well below the breakdown voltage (For thin films, the voltage range may be high). Previous studies^{13,14} demonstrated that even kV-scale voltage can be applied to single crystal ferroelectric LiNbO₃ surface.

We additionally conducted a similar experiment with high voltage application case but this time we performed the experiment at a moderated voltage range. Fig. R8 depicts the friction and topography images during the milling scan while bias application from -10 V to 10 V. Even with the moderate voltage range, we observe no significant difference depending on the applied bias in both up and down domains, however we do observe some friction signal change in the same voltage regime possibly because of the complex interaction of probe with wear debris or adsorbates.

Last, the milling experiment has already been tested using 8 different types of probes, which including non-conductive, weakly conductive and highly conductive probes. As shown in Table R2, the asymmetric tribology trend is not dependent on the conductivity and substance of probe, but rather on the spring constant of the probe.

Fig. R8. Electrical bias application during the milling scan. (a) Friction during the milling scan at optimized loading force with application of bias from -10 V to 10 V. **(b)** Height measured during the milling scans (scan angle of 90° with the fast scan axis perpendicular to the domain walls).

Comment 2: Another possible effect is that there will always be a contact potential difference between the tip and sample (whether the tip is conductive or not). Taking the tip's work function

as fixed, the contact potential will be different between up and down domains. This again will lead to different total normal loads, higher pressures, higher friction, and more wear.

The authors exclude an electrostatic switching effect by running tests at voltages ranging from -150 to +150 volts, but this is an abnormally large voltage to apply to an AFM tip. This often can cause dielectric breakdown, large polarization forces, and other difficult-to-control effects. The authors should show what happens at more modest voltages, like -10 to +10 V, where they are likely to cross over the contact potential difference between the tip and sample. They can also measure or at least estimate the contact potential difference with Kelvin Probe force microscopy, or simply by measuring the deflection of the tip as the voltage is ramped from -10 to +10 V while the tip hovers close to, but not in contact with, the surface.

Response 2: We value the reviewer's suggesting alternative mechanism to discuss. Regarding experiments involving high voltage and moderate voltage application, please refer to the response #1. We believe that we already appropriately responded to the comment. Even in the high/moderate voltage range which likely cross over the contact potential difference between the tip and sample, no significant tribology difference was observed. Further, in response to reviewer, we also conducted KPFM measurement on pristine/worn PPLN surface as shown in Fig. R9. We could not observe a significant difference in contact potential between pristine and worn PPLN surface, however we do observe the potential difference between up and down domains even after 10 hours of milling scan. As mentioned in the manuscript, if the screening condition is relevant to the observed asymmetry, charge injection during the scan (from -150 V to 150 V, or -10 V to 10 V in the moderate manner) should be effective, however, there is no significant change in the asymmetrical tribology behavior.

Fig. R9. (a) Height, (b) KPFM potential of pristine and worn PPLN surface. (c) Reference PFM phase image shows polarization configuration. KPFM was conducted after 10 hours following the milling scan. NM-TC (*Adama Innovations*) was used for both milling and KPFM scan.

Comment 3: They could also have checked for differential adhesion by measuring the pull-off (adhesion) force between the tip and sample. This is a very routine measurement. Maps of adhesion over the different domains is thus necessary for this paper. The authors would also benefit from measuring the friction force as a function of load, down to the pull-off force, which can give a better sense of the contact mechanics at play.

Response 3: We agree with the reviewer. We carried out additional adhesion force measurements (Fig. R10). Adhesion force maps depending on loading forces, polarization direction on pristine

and worn PPLN surface are shown in Fig. R10 with height and PFM phase images. The adhesion forces were measured at the boundary between pristine and worn PPLN surface. As can be seen in Figs. R10c–e, no discernible difference in adhesion force was observed in pristine and milled ferroelectric surface. At lower force regime (e.g., in Fig. R10 from 100 nN to 400 nN), therefore, there is no significant adhesion force difference depending on the polarization direction, and again, the mechanism for the polarization-dependent tribology is triggered by the application of high strain gradient.

Fig. R10. Adhesion force mapping on pristine and worn PPLN surface. (a) Topography and (b) PFM phase of pristine and work PPLN surface. (c–e) Adhesion force mapping with loading force of 100 nN (c), 200 nN (d) and 400 nN (e). A NM-TC probe (*Adama innovations*) was used for milling, height, PFM imaging and adhesion mapping.

Comment 4: On other possible aspect to consider is the role of mechanochemistry in wear. Stress-induced acceleration of wear through an Eyring-like model, based on Arrhenius kinetics, has been

demonstrated in several cases with AFM. The idea is that atomic-scale wear involves breaking a bond (or bonds) between the atom being removed and the other atoms in the solid to which it was coordinated. This is a chemical process, and Eyring and Bell, showed that this can be described by a reduction of the energy barrier for the bond breaking processes that depends linearly on the applied force (Bell model) or stress (Eyring model). Could an electric field be coupled in to this framework? Perhaps something could be determined by analogy to electric field effects in chemical reactions.

Response 4: We appreciate the reviewer's detailed feedback. We believe that the response #5 for reviewer 1 appropriately address this comment.

Comment 5: Regarding my second major concern (unclear explanation of their model): The authors do not explain the essential physical mechanism underlying their explanation for the observed effect. They say in the manuscript “flexoelectrically induced polarization and the consequent asymmetry of mechanical properties between oppositely polarized up and down domains.” What mechanical properties are they referring to? Elastic moduli? Yield strengths? Fracture toughnesses? Moreover, the statement above is not clear or specific; the details are left to a model described mostly in Methods and Supplementary Text, which is difficult to follow and filled with mechanics jargon. The authors need to clearly lay out what the mechanism is.

Response 5: Please see responses #1 and #5 for reviewer 1. We think that it adequately addresses the reviewer's concern of the mechanism description. We have also updated the manuscript with additional explanation, and simplified the supplementary information explaining the theory. Regarding the sentence quoted by the referee, we revised the sentence like below to improve comprehension. We sincerely thank the reviewer.

Before: Taken together, our simulation and experimental observations including other possibilities described in Supplementary Information suggest that the dominant mechanism at the origin of the asymmetrical tribology is flexoelectrically induced polarization and the consequent asymmetry of mechanical properties between oppositely polarized up and down domains.

After: Taken together, our simulation and experimental observations including other possibilities described in Supplementary Information suggest that the dominant mechanism at the origin of the asymmetrical tribology is flexoelectrically induced polarization and the consequent asymmetry of mechanical **responses** between oppositely polarized up and down domains.

Comment 6: They also share – to their credit – the unusual observation that the friction contrast changes depending on the slow-scan direction of the AFM image. This is very unusual, and their explanation is that “the height and friction differences signals (Fig. 1h) oscillate because of the contact geometry difference between frame-up (scan from below to above) and frame-down (scan from above to below) during the continuous, repeated milling scans.” This doesn’t cut it. What evidence do they have that the contact geometry changes with the slow scan direction? Why would it? Does the magnitude and sign of this difference change if different tips are used? It is likely that something else is going on, and it may be a clue regarding the mechanism at play, perhaps related to the displacement of adsorbates on the surface by the scanning action tip, or perhaps cross-talk between normal and lateral signals of the cantilever/photodiode. I don’t have an explanation, but it’s odd, and the effect is strong. Note that only the friction signal oscillates; the height differences does not, although the etch rate (derivative of height with scan number) does appear to oscillate.

Response 6: We appreciate the reviewer’s keen comment. Thanks to the reviewer, we first reexamined all the milling scan data used in the Fig. 1, and discovered a very interesting fact. The acquired milling images were not continuous 50 scans, but intermediate PFM scans were conducted after 10th, 20th, and 30th milling scan to check domain stability. Even though the PFM scans were performed at much lower loading force (approximately ten times lower than milling force), the height and friction difference trends significantly change as shown in Fig. R11. This result also helps us to understand the mechanism, as the complex attachment of wear debris or adsorbates significantly changes friction and wear trend during the low-force scanning with alternating electrical bias (3V). These findings suggest that the tribological interactions between the probe and crystal are highly sensitive, and can be controlled by manipulating the probe-sample interface.

Thus, we change our sentence structures like below.

Before: The height and friction differences are clearly correlated with increasing scan number and show three distinct regimes. In region 1, the height difference slightly increases with relatively stable friction difference. In region 2, it drastically increases with higher friction difference, and the slope of height difference (etch rate) approaches a maximum value with maximum friction difference. In region 3, the height difference finally starts to saturate with constant lower friction difference.

After: Further, the slope of height difference (relative wear rate) is clearly correlated with the friction difference, and the trend shows four distinct regimes in Fig. 1h. Interestingly, the boundaries between each regime correspond to where we performed intermediate PFM scans (indicated by *). In particular, to check the domain stability after 10th, 20th and 30th milling scans, we conducted PFM scans during which we apply approximately one tenth of the loading force, as compared to milling scans. After each of these intermediate PFM scans, height and friction difference trends significantly change. These findings suggest that the tribological interactions between the probe and the ferroelectric are highly sensitive, and can be controlled by manipulating the probe-sample interface.

Fig. R11. Height and friction difference between up and down domains with increasing scan number in Fig. 1. After 10th, 20th and 30th milling scans (indicated by *), intermediate PFM scans were conducted.

Regarding oscillation of friction and height differences in our manuscript, we keep our claim that the relative contact geometry difference between frame-up and frame down scans can affect the observed oscillating behavior in Fig. 1h. In such relative contact geometry can be substantially

changed by either tip geometry or complex interaction among tip, sample and surface particles. Sometimes, the geometry difference can originate from the nature of cantilever buckling during the high force contact scans. We explain the oscillation of height and friction differences as follows.

1) First, we observe the oscillating behavior not only in the frictional difference between up and down domains but also in the global evolution of the height and vertical deflection, friction of the measurement series (shown for the last 10 scans of the series in Fig. R12, averaged over the whole image), with a consistent oscillation in the values of the measurables. We believe that this global oscillation during the 50 scans is due to the continuous nature of the force applied along the long axis of the cantilever due to the slow scan axis. During a frame-down scan, the cantilever is dragged, whereas during the frame-up scan, it is continuously pushed - both resulting in cantilever buckling and therefore geometry changes (consistent with the observed oscillations in the height, vertical deflection and friction). Because of this global oscillation caused by a change in contact geometry, the height and friction differences in Fig. 1h oscillate as well. However, we observe the consistent trend; friction and wear rate are always greater in ferroelectric up domains.

Fig. R12. Global evolution of the height, vertical deflection and friction signal during last 10 scans in Fig. 1. We note that the each point in graph consists of averaged height, deflection and friction data from the whole image, which differs from the difference between up and down domains in Fig. 1h.

2) As described by the previous discussion in the same response, the trend of friction and wear is highly sensitive and can be changed after intermediate PFM scans. Specifically, relative contact geometry (either tip geometry or the interposition between probe and surface) can significantly alter the friction and wear trend. In fact, the tip used for this measurement (NM-TC, *Adama innovations*) exhibits an asymmetric indentation profile as shown in Fig. R13, where the tip was driven into the surface with a maximum force of 40 μN - and thus we would expect such a geometry to affect the oscillations strongly when the tip is dragged up or down along the slow scan axis.

Fig. R13. Height image acquired after 8×8 indentations with a maximum loading force of $40 \mu\text{N}$ using NM-TC on PPLN.

3) Lastly, we have observed this phenomenon in other types of measurements on various materials and under different conditions, but have usually discarded the effects as artifacts. Although it may be an interesting phenomenon to investigate in a separate study on the cantilever dynamics/artifacts under applied loads, we do not believe that it enters in the scope of this study.

Comment 7: Finally, going back to the comment above about the “asymmetry of mechanical properties”: the authors themselves (to their credit) note that “this asymmetry implies an anomalous, positive correlation between the hardness of the domains and their etch rate, while normally high hardness materials are more resistant to wear as reported in the form of Ashby plot between hardness and wear coefficient.” It’s honest of them to point this out, but doesn’t it suggest that the model may not be correct?

Response 7: We appreciate the keen and insightful comment. Mechanical wear is a complex phenomenon, but shows a general trend of negative correlation between hardness and wear coefficient¹⁶. In other words, the larger the hardness, the smaller the wear coefficient. In our case, this general trend is broken due to the complicated interactions between external screening charges and flexoelectrically coupled polarization. From the previous study, ferroelectric up domains are harder than down domains¹⁷. However, we were able to preferably wear the high hardness domains (up domains) by finding the specific conditions which suppress external screening charge effects and enhance the flexoelectrically coupled friction at the nanoscale or even sub-nm scale precision. By definition from Oliver-Pharr model, hardness is the maximum load divided by the contact area.

Under a fixed load, the contact area and indentation depth are directly related to the hardness of the material. Here, we claim that contact area or indentation depth of ferroelectric materials can be controlled by adjusting the strain gradient.

In addition to asymmetric milling experiments with a diamond AFM tip, we conducted nanoindentation experiments to further understand the anomaly and found that there are two distinct mechanisms of asymmetric response depending on the applied load. More specifically, strain gradient to the sample can be controlled by manipulating loading forces through a tip or changing the tip geometry. We conducted two different experiments; One is nanoindentation measurements using two different tips (spherical and Berkovich indenters), the other is AFM tribology experiments with different loading forces. Loading curves from the nanoindentation measurements allows us to observe indentation depth as a function of tip geometry and load. The wear depth depending on the loading forces can be obtained from the milling experiments using the AFM tip (Fig. S27).

The results from two different experiments exhibit a consistent trend and are summarized in Tables R3 and R4. Under higher load, indentation depth or wear depth is greater in ferroelectric down domains, but indentation depth or wear depth is higher in ferroelectric up domains under moderate load. In optimal strain gradient regime (e.g., moderate load application with spherical probe or with the diamond AFM tip (5 or 10 μN case in Fig. S7), blue color in Tables R3 and R4), the suggested model in Fig. 2 is valid. Indentation depth and contact area is greater in ferroelectric up domains, and therefore frictional responses and wear depth are also greater in ferroelectric up domains.

In high strain gradient regime (e.g., high load application with Berkovich probe or high loading force application using diamond AFM tip (20 μN case in Fig. S7), red color in Table R4), ferroelectric down domains show higher indentation depth and wear depth. This is maybe due to pronounced crack propagation in ferroelectric down domains under significantly high load. In this regime, because of pronounced crack propagation in down domains, indentation depth is higher in ferroelectric down domains, and the asymmetry trend become opposite. It thus leads higher wear depth in ferroelectric down domains. This insight is also match with previous study on crack propagation of ferroelectrics, where the crack length is longer along the ferroelectric polarization².

In summary, ferroelectrics have asymmetric mechanical and tribological properties depending on the polarization orientation. Furthermore, even in the same polarization direction, nanomechanical and tribological behavior is controllable by adjusting the amount of strain gradient

or stress to the crystals. We plan to do further systematic study on these controllable mechanical responses of ferroelectrics in the future, but the detailed study is beyond the scope of this manuscript.

Type of indenter	Load (mN)	Indentation depth (Ferroelectric up, nm)	Indentation depth (Ferroelectric down, nm)
Spherical	2	20.025 ± 1.920	15.807 ± 0.999
Berkovich	2	76.082 ± 1.171	77.645 ± 0.836
Berkovich	5	138.438 ± 1.185	140.629 ± 1.080
Berkovich	10	211.990 ± 2.010	214.488 ± 2.036

Table R3. Indentation depth from loading curves in nanoindentation measurement using two different types of indenter. 15 data points are averaged in each case. Blue color shows results under moderate load, where ferroelectric up domains have higher indentation depth, and red color shows results under high load, where ferroelectric down domains show higher indentation depth.

Type of AFM tip	Load (μN)	Wear depth (Ferroelectric up, nm)	Wear depth (Ferroelectric down, nm)
Single crystal diamond	5	2.019 ± 0.171	1.662 ± 0.132
Single crystal diamond	10	13.035 ± 0.730	10.583 ± 0.517
Single crystal diamond	20	104.510 ± 5.585	114.492 ± 7.083

Table R4. Averaged wear depth from Fig. S7. Blue color shows results under moderate load, where ferroelectric up domains have higher wear depth, and red color shows results under high load, where ferroelectric down domains show higher wear depth.

Comment 8: In short, the authors need to shore up their and better explain model, consider other possible mechanisms (including ideas presented in literature that were not cited), and explain the unusual slow-scan direction observation.

Response 8: We appreciate the reviewer’s comments to this point, and we are confident that we have addressed all the concerns from the reviewer.

Comment 9: Finally, a few minor observations: Abstract: I don’t like “Artificially induced” since it’s not artificial; perhaps they mean “Externally induced” or just “Switchable”

Response 9: Thank you very much, we modified the phrase from “Artificially induced” to “Switchable”.

Before: Artificially induced asymmetric tribological properties of ferroelectrics offer an alternative route to visualize and control ferroelectric domains.

After: **Switchable** tribological properties of ferroelectrics offer an alternative route to visualize and control ferroelectric domains.

Comment 10: Abstract: the fact that a diamond tip was used should be mentioned; wear depends on both materials in contact, not just one.

Response 10: We appreciate the reviewer's comment, but we also observed the asymmetric wear behavior using Pt/Ti-coated Si probe (Fig. R4), we believe that various types of probe with high spring constant could be used. As such, we keep our statement like below.

Before and after: Here, we observe the switchable friction and wear behavior of ferroelectrics using a nanoscale scanning probe where down domains having lower friction coefficient than up domains can be used as smart masks as they show slower wear rate than up domains.

Comment 11: Abstract: the last sentence seems a bit exaggerated, especially the use of "establish".

Response 11: Thank you very much, we modified the phrase from "establish" to "demonstrate" in response to the reviewer's comment.

Before: These findings establish that ferroelectrics are electrically tunable tribological materials at the nanoscale for versatile applications.

After: These findings **demonstrate** that ferroelectrics are electrically tunable tribological materials at the nanoscale for versatile applications.

Comment 12: Lines 46-47 "friction and wear coefficients" are empirical quantities. "Shear strength" can just be used instead of "friction coefficient" for example. Wear is trickier but I don't believe the "wear coefficient" is something they are controlling.

Response 12: We cherish the reviewer's comment, and we updated the phrase from "friction and wear coefficients" to "friction and wear behavior". We think that we can possibly manipulate the surface friction and wear behavior by switching the ferroelectric polarization.

Before: As we shall show in this article, increasing this flexoelectric contribution under highly inhomogeneous stress also has emergent consequences for coupled tribological properties—in particular, friction and wear coefficients—which in turn can be exploited for direct visualization of ferroelectric domains and extremely fine physical lithography without the need for masks or chemical reagents.

After: As we shall show in this article, increasing this flexoelectric contribution under highly inhomogeneous stress also has emergent consequences for coupled tribological properties—in particular, **friction and wear behavior**—which in turn can be exploited for direct visualization of ferroelectric domains and extremely fine physical lithography without the need for masks or chemical reagents.

Comment 13: Lines 50-51: "...no previous studies..." This is a rather sweeping statement. Are the authors sure? The final sentence of the same paragraph is more measured.

Response 13: We thank the reviewer's feedback, and have removed the statement from the text.

Comment 14: Line 64: I believe the authors mean to say "... is asymmetric and *the existence of the asymmetry* is independent of surface chemistry" since the magnitude of the asymmetry surely varies with the surface chemistry.

Response 14: We appreciate the reviewer's keen advice. We modified the sentence as follows based on the reviewer's comment.

Before: We demonstrate that the local friction and wear behavior of ferroelectrics is asymmetric and independent of surface chemistry under large strain gradient, and that this inherent tribological asymmetry enables facile and reversible control of friction and wear properties, which can be exploited for nano-lithographic patterning by simply "rubbing" the surface of a voltage-written ferroelectric.

After: We demonstrate that the local friction and wear behavior of ferroelectrics is asymmetric and **the existence of the asymmetry is independent of surface chemistry** under large strain gradient, and that this inherent tribological asymmetry enables facile and reversible control of friction and wear properties, which can be exploited for nano-lithographic patterning by simply “rubbing” the surface of a voltage-written ferroelectric.

Comment 15: Line 91: “polarization-dependent tribology”. Since tribology means friction, wear, adhesion, and more, what specifically are the authors referring to here? I think they mean to say “polarization-dependent friction and wear”

Response 15: We appreciate the reviewer’s comment. We updated the phrase from “polarization-dependent tribology” to “polarization-dependent friction and wear” in response to the reviewer’s opinion.

Before: We observed polarization-dependent tribology of the sample surface using single-crystalline conductive diamond probes (NM-TC, *Adama Innovations*), selected for their extreme hardness and stiffness, with relatively high contact 94 loading force (5 μN) and scan rate (4.88 Hz, equivalently 146.48 $\mu\text{m/s}$).

After: We observed **polarization-dependent friction and wear** of the sample surface using single-crystalline conductive diamond probes (NM-TC, *Adama Innovations*), selected for their extreme hardness and stiffness, with relatively high contact loading force (5 μN) and scan rate (4.88 Hz, equivalently 146.48 $\mu\text{m/s}$).

Comment 16: Line 170: “Because friction strongly depends on the real contact area, as does the mechanical wear rate,” seems off. It would make more sense to say that, at higher applied normal stresses (due to higher normal load), one gets more contact area (and thus friction force), and one gets more wear because of higher normal stresses (and perhaps also because of the higher friction forces).”

Response 16: We value the reviewer’s input. We changed the sentence structure as follows in response to the reviewer’s remark.

Before: Because friction strongly depends on the real contact area²⁸, as does the mechanical wear rate, we consequently expect higher friction and wear in up domains than down domains (Fig. 2b,c).

After: Because applied stress at larger contact area region could induce higher friction, and thus more wear²⁸, we consequently expect higher friction and wear in up domains than down domains (Fig. 2b,c).

References

1. Abdollahi, A. *et al.* Fracture toughening and toughness asymmetry induced by flexoelectricity. *Phys. Rev. B* **92**, 094101 (2015).
2. Cordero-Edwards, K., Kianirad, H., Canalias, C., Sort, J. & Catalan, G. Flexoelectric fracture-ratchet effect in ferroelectrics. *Phys. Rev. Lett.* **122**, 135502 (2019).
3. Long, C. J., Ebeling, D., Solares, S. D. & Cannara, R. J. Friction imprint effect in mechanically cleaved BaTiO₃ (001). *J. Appl. Phys.* **116**, 124107 (2014).
4. Liao, Y. & Marks, L. D. Direct observation of layer-by-layer wear. *Tribol. Lett.* **59**, 1-11 (2015).
5. Liao, Y. & Marks, L. In situ single asperity wear at the nanometre scale. *Int. Mater. Rev.* **62**, 99-115 (2017).
6. Jacobs, T. D. & Carpick, R. W. Nanoscale wear as a stress-assisted chemical reaction. *Nat. Nanotechnol.* **8**, 108-112 (2013).
7. Ștoflea, L. E., Apostol, N. G., Trupină, L. & Teodorescu, C. M. Selective adsorption of contaminants on Pb (Zr, Ti)O₃ surfaces shown by X-ray photoelectron spectroscopy. *J. Mater. Chem. A* **2**, 14386-14392 (2014).
8. Mizzi, C. A., Lin, A. Y. & Marks, L. D. Does flexoelectricity drive triboelectricity? *Phys. Rev. Lett.* **123**, 116103 (2019).
9. Mizzi, C. A. & Marks, L. D. When flexoelectricity drives triboelectricity. *Nano Lett.* **22**, 3939-3945 (2022).

10. Olson, K. P., Mizzi, C. A. & Marks, L. D. Band bending and ratcheting explain triboelectricity in a flexoelectric contact diode. *Nano Lett.* **22**, 3914-3921 (2022).
11. Gellé, F. *et al.* Guideline to atomically flat TiO₂-terminated SrTiO₃ (001) surfaces. *Surf. Sci.* **677**, 39-45 (2018).
12. Park, S. M. *et al.* Colossal flexoresistance in dielectrics. *Nat. Commun.* **11**, 2586 (2020).
13. Wang, L. *et al.* Flexoelectronics of centrosymmetric semiconductors. *Nat. Nanotechnol.* **15**, 661-667 (2020).
14. Rosenman, G., Urenski, P., Agronin, A., Rosenwaks, Y. & Molotskii, M. Submicron ferroelectric domain structures tailored by high-voltage scanning probe microscopy. *Appl. Phys. Lett.* **82**, 103-105 (2003).
15. Agronin, A., Rosenwaks, Y. & Rosenman, G. Direct observation of pinning centers in ferroelectrics. *Appl. Phys. Lett.* **88** (2006).
16. Ashby, M. F. & Cebon, D. Materials selection in mechanical design. *MRS Bull.* **30**, 995 (2005).
17. Cordero-Edwards, K., Domingo, N., Abdollahi, A., Sort, J. & Catalan, G. Ferroelectrics as smart mechanical materials. *Adv. Mater.* **29**, 1702210 (2017).

REVIEWER COMMENTS

Reviewer #1 (Remarks to the Author):

The authors have made many changes which improve the manuscript. I still think that splitting would have made sense, but if the editors are OK with such a massive SM then I am as well. I will caution the authors that the main ones should do a little more reading into friction. For instance, they argue that fracture toughness does not matter because the surfaces smooth, which misses the point; the empirical connection between wear and fracture toughness is well known, smoothing is different.

Reviewer #2 (Remarks to the Author):

This manuscript makes an excellent point, artificially induced asymmetric tribological properties of ferroelectrics, that offers an alternative route to visualize and control ferroelectric domains. The switchable friction and wear behavior of ferroelectrics using a nanoscale scanning probe can be used as smart masks, and the asymmetry is enabled by flexoelectricity coupled polarization in the up and down domains under a sufficiently high contact force. The polarization-sensitive tribological asymmetry is widely applicable across various ferroelectrics with different chemical composition and crystalline symmetry. And using this switchable tribology and multi-pass patterning with a domain-based dynamic smart mask, this manuscript demonstrated three-dimensional nanostructuring exploiting the asymmetry wear rates of up and down domains. The findings establish that ferroelectrics are electrically tunable tribological materials at the nanoscale for versatile applications. There is one manuscript, "Flexoelectricity in periodically poled lithium niobate, Journal of Physics D: Applied Physics, 55 335303, 2022", might be helpful of illustrating the mechanism. The reviewer recommends its publication on Nature Communications.

Reviewer #3 (Remarks to the Author):

I am generally satisfied with the authors' responses which are comprehensive and substantial. My remaining concern is that the mechanism of wear still remains poorly explained, even in the form of a hypothesis. The description on pages 7-8 (lines 194-201) is still vague. The authors write, "Since flexoelectric polarization

interacts differently with up and down domains..." (how? why?) "... this electric potential distribution varies depending on the direction of out-of-plane polarization." (how? why?) "...Piezoelectric and flexoelectric polarization fields upon indentation (Supplementary Fig. 19) thus lead to different competing or synergistic combinations in up and down domains". How? Why? What competition? What synergy? Please explain without jargon, for a physical science audience who are not experts in flexoelectricity. If they can do this, the paper is suitable for publication.

Response to Reviewers

For convenience of the editor and referees, comments provided by reviewers are presented in blue and our responses are inserted in black. In the manuscript, the modifications are highlighted in yellow.

Reviewer 1

Comment: The authors have made many changes which improve the manuscript. I still think that splitting would have made sense, but if the editors are OK with such a massive SM then I am as well. I will caution the authors that the main ones should do a little more reading into friction. For instance, they argue that fracture toughness does not matter because the surfaces smooth, which misses the point; the empirical connection between wear and fracture toughness is well known, smoothing is different.

Response: We express our gratitude for the valuable feedback provided by the reviewer. Regarding the reviewer's comment on the surface smoothing and wear, we regret the manner in which we initially expressed the paragraph. We fully agree with the reviewer that fracture toughness has strong connection with wear properties. Indeed, at a significantly high strain gradient regime, our experimental observations also follow the general empirical relation between fracture toughness and wear rate^{1,2}, where greater fracture toughness corresponds to lower wear rates. In this regime, high hardness domains (up domains) show greater fracture toughness, and also show lower indentation depth and wear depth (20 μN case in Fig. R1, Fig. R2a–b and red box in Table R1–2).

However, in the optimized strain gradient regime, this general trend is broken due to the complicated interactions between external screening charges and flexoelectrically coupled polarization (5–10 μN case in Fig. R1, Fig. R2c and blue box in Table R1–2). In this regime, we were able to preferably wear high hardness domains (up domains) by finding the specific conditions which suppress external screening charge effects and enhance the flexoelectrically coupled friction at the nanoscale or even sub-nm scale precision.

Furthermore, we observed the roughening of surface in the significant high strain gradient regime (20 μN case in Fig. R1c), and surface smoothing in optimized regime (Fig. R3). We also

have presented asymmetric mechanical and tribological behaviors in two regimes showing different resulting roughness. That is why we initially presented that fracture toughness is not dominant in the optimized regime in the response letter, because it does not follow the general trend. Our finding suggests that even in the same polarization direction, nanomechanical and tribological behaviors are controllable by adjusting the amount of strain gradient or stress to the crystals.

Fig. R1. Contact force optimization of asymmetric friction and wear in PPLN. (a) Height image after ten milling scans with increasing loading forces in four different regions. (b) PFM phase image after ten milling scans. (c) Height in each region with different loading forces from 2.5 to 20 μN. (d) Wear depth as a function of loading force in each region. (e) Wear depth as a function of loading force except for the failure case. Orange color is for up domains and purple color is for down domains. We note that the wear depth is higher in up domains up to 10 μN, however it is lower in up domains at 20 μN. More details can be found in Fig. S7.

Type of AFM tip	Load (μN)	Wear depth (Ferroelectric up, nm)	Wear depth (Ferroelectric down, nm)
Single crystal diamond	5	2.019 ± 0.171	1.662 ± 0.132
Single crystal diamond	10	13.035 ± 0.730	10.583 ± 0.517
Single crystal diamond	20	104.510 ± 5.585	114.492 ± 7.083

Table R1. Average wear depth from Fig. R1. Blue color shows results under moderate load, where ferroelectric up domains have higher wear depth, and red color shows results under high load, where ferroelectric down domains show higher wear depth.

Fig. R2. Nanoindentation of stoichiometric LiNbO₃ single crystals. (a) Hardness and (b) load-depth curve of ferroelectric up and down crystals from the Berkovich nanoindentation measurements. Maximum load is 10 mN. Ferroelectric down crystal shows higher hardness, so lower indentation depth than up crystal. (c) Loading and unloading load-depth curve using spherical nanoindenter in ferroelectric up and down crystals. Ferroelectric up crystal shows higher indentation depth than down crystal. Maximum load is 3 mN. Average values of indentation depth in each case are shown in Table R2. More details can be found in S10.

Type of indenter	Load (mN)	Indentation depth (Ferroelectric up, nm)	Indentation depth (Ferroelectric down, nm)
Spherical	2	20.025 ± 1.920	15.807 ± 0.999
Berkovich	2	76.082 ± 1.171	77.645 ± 0.836
Berkovich	5	138.438 ± 1.185	140.629 ± 1.080
Berkovich	10	211.990 ± 2.010	214.488 ± 2.036

Table R2. Indentation depth from loading curves in nanoindentation measurement using two different types of indenter. 15 data points are averaged in each case. Blue color shows results under moderate load, where ferroelectric up domains have higher indentation depth, and red color shows results under high load, where ferroelectric down domains show higher indentation depth. More details can be found in S10.

Fig. R3. Surface roughness before (a) and after (b) milling scans. Orange color represents the region acquired for the roughness in up domains, while purple indicates the region for the roughness in down domains. The roughness decreases after milling scans in both polarization domains.

Reviewer #2

This manuscript makes an excellent point, artificially induced asymmetric tribological properties of ferroelectrics, that offers an alternative route to visualize and control ferroelectric domains. The switchable friction and wear behavior of ferroelectrics using a nanoscale scanning probe can be used as smart masks, and the asymmetry is enabled by flexoelectricity coupled polarization in the up and down domains under a sufficiently high contact force. The polarization-sensitive tribological asymmetry is widely applicable across various ferroelectrics with different chemical composition and crystalline symmetry. And using this switchable tribology and multi-pass patterning with a domain-based dynamic smart mask, this manuscript demonstrated three-dimensional nanostructuring exploiting the asymmetry wear rates of up and down domains. The findings establish that ferroelectrics are electrically tunable tribological materials at the nanoscale for versatile applications. There is one manuscript, “Flexoelectricity in periodically poled lithium niobate, *Journal of Physics D: Applied Physics*, 55 335303, 2022”, might be helpful of illustrating the mechanism. The reviewer recommends its publication on *Nature Communications*.

Response: We appreciate time and effort of the reviewer’s for evaluating our manuscript. We added the reference suggested by the reviewer in the introduction part (Reference 11).

Reviewer #3

I am generally satisfied with the authors' responses which are comprehensive and substantial. My remaining concern is that the mechanism of wear still remains poorly explained, even in the form of a hypothesis. The description on pages 7-8 (lines 194-201) is still vague. The authors write, "Since flexoelectric polarization interacts differently with up and down domains..." (how? why?) "... this electric potential distribution varies depending on the direction of out-of-plane polarization." (how? why?) "...Piezoelectric and flexoelectric polarization fields upon indentation (Supplementary Fig. 19) thus lead to different competing or synergistic combinations in up and down domains". How? Why? What competition? What synergy? Please explain without jargon, for a physical science audience who are not experts in flexoelectricity. If they can do this, the paper is suitable for publication.

Response: We appreciate the positive comment from the reviewer and we additionally tried to explain the mechanism better in response to the concern. Flexoelectric polarization arises as a result of the inversion symmetry breaking which automatically occurs whenever any material is subjected to a strain gradient. Unlike piezoelectricity or ferroelectricity, which are characterized by inherent polar order (reversible under applied electric field, for the latter) and thus present in only certain crystal groups, flexoelectric polarization can therefore be induced in essentially any material. However, it leads to a noticeable effect only when the flexoelectric coefficient coupling the polarization and strain gradient is relatively large, as is the case for materials which are also independently ferroelectric. While this phenomenon has been known for decades³, measurements on bulk ferroelectric crystals or ceramics showed relatively small flexoelectric polarization, since their relatively macroscopic size meant that under an application of force the resulting strain gradient remained modest.

It was only relatively recently that the combination of epitaxial ferroelectric thin films and scanning probe microscopy allowing extremely local application of mechanical pressure, and thus a very high strain gradients, led to significant values of the flexoelectric polarization. What is crucial in this geometry is that while the ferroelectric polarization can be a priori oriented UP or DOWN with respect to the out-of-plane polar axis of the materials under consideration, the flexoelectric polarization, resulting from the SPM tip pressing down onto the surface, only arises in one specific orientation. The two can thus either act synergistically, if their orientations coincide,

resulting in a larger effective total polarization under the tip during force application, or they can compete. Pioneering work in this field by the groups of Catalan and Gruverman on BaTiO₃ thin film showed that the resulting flexoelectric polarization can in fact be comparable or even greater than the purely ferroelectric polarization, and thus demonstrated mechanical switching of the ferroelectric polarization⁴. The effects of strain gradients due to lattice mismatch during sample growth, while smaller than those which can be induced via SPM, can also play a role, as demonstrated by the Noh group on BiFeO₃⁵.

Subsequent questions turned to the mechanical effects of this flexoelectric polarization. Specifically, while the contact stiffness between a probe and a material depends on the intensity of the polarization, and thus should be equivalent for UP and DOWN domains when only the ferroelectric contribution is considered, this is no longer the case when the contribution of the flexoelectric polarization, acting with or against the ferroelectric one, is taken into account. Effectively, under the application of strain gradient with a classic tribology probe or more locally the SPM tip the difference in total polarization for the UP and DOWN domains results in an asymmetry of the measured mechanical properties. This asymmetry in terms of stiffness, crack propagation length, and contact resonance response was beautifully demonstrated in a series of recent papers by the Catalan group^{6,7}.

Given the high impact of these publications, and their wide dissemination in popular science news⁸ (the catch phrase of “Turning a material upside-down can sometimes make it softer” was clearly very attractive), we did not go into a detailed explanation of this phenomenon in our manuscript, rather referring the reader to this extensive body of work.

Following the referee’s comments, we have modified the text to include a very brief explanation. We chose not to include the above detailed text in the SM as these are already ample, and we believe that the interested reader can be better served by going directly to the beautiful articles of the Catalan group^{6,7}.

The revised version includes the text to clarify the mechanism we propose as follows.

Page 7-8: "Indentation with the spherical diamond probe generates a spreading electric potential distribution in the LiNbO₃ crystal (Supplementary Fig. 18). Since flexoelectric polarization always results from downward pressure-induced strain gradients, and thus interacts differently with up and down oriented ferroelectric domains, this electric potential distribution varies depending on the direction of out-of-plane polarization."

Additional minor modifications

- 1) We have additionally changed some typos in the revised manuscript and the modifications are marked in yellow.
- 2) We also have changed typo in the young's modulus value used in simulation from 103 GPa to 146 GPa. The modified value is based on the least squares method⁹ from the tensor values in Ref. ¹⁰.
- 3) We have further modified the schematic in Fig. 2d. The concept schematic was first presented as shown in Fig. R4a, since the potential difference of complete screening case is negligible compared to the scraped case¹⁰. However, we have changed the schematic because there is an actual potential change between screened up and down domains, as we also observed the KPFM potential difference (Fig. R5). The previous study¹¹ also provides support for the redesigned scheme.

Fig. R4. Modification of Fig. 2d. (a) Schematic before modification, (b) schematic after modification.

Fig. R5. (a) Height, (b) KPFM potential of pristine and worn PPLN surface. (c) Reference PFM phase image shows polarization configuration. KPFM was conducted after 10 hours following the milling scan. NM-TC (*Adama Innovations*) was used for both milling and KPFM scan.

References

1. M. F. Ashby, D. Cebon, Materials selection in mechanical design. *MRS Bull.* **30**, 995 (2005).
2. Ding, Z., Zhou, S. & Zhao, Y. Hardness and fracture toughness of brittle materials: A density functional theory study. *Phys. Rev. B* **70**, 184117 (2004).
3. Kogan, S. M. Piezoelectric effect during inhomogeneous deformation and acoustic scattering of carriers in crystals. *Sov. Phys.-Solid State* **5**, 2069-2070 (1964).
4. Lu, H. *et al.* Mechanical writing of ferroelectric polarization. *Science* **336**, 59-61 (2012).
5. Lee, D. *et al.* Giant flexoelectric effect in ferroelectric epitaxial thin films. *Phys. Rev. Lett.* **107**, 057602 (2011).
6. Cordero-Edwards, K., Domingo, N., Abdollahi, A., Sort, J. & Catalan, G. Ferroelectrics as smart mechanical materials. *Adv. Mater.* **29**, 1702210 (2017).
7. Cordero-Edwards, K., Kianirad, H., Canalias, C., Sort, J. & Catalan, G. Flexoelectric fracture-ratchet effect in ferroelectrics. *Phys. Rev. Lett.* **122**, 135502 (2019).
8. Autonomous University of Barcelona, Turning a material upside-down can sometimes make it softer, Phys.org https://phys.org/news/2017-10-material-upside-down-softer.html#google_vignette/ (2017)
9. Jiang, B.-N. On the least-squares method. *Comput. Methods Appl. Mech. Eng.* **152**, 239-257 (1998).
10. Persson, K. *Materials Data on LiNbO₃ (SG:161) by Materials Project* (2014).
11. Kalinin, S. V. & Bonnell, D. A. Local potential and polarization screening on ferroelectric surfaces. *Phys. Rev. B* **63**, 125411 (2001).